# Planktonic Aggregation Enhances Antibiotic Tolerance in Non-MDR *Acinetobacter baumannii*

**DOI:** 10.3390/microorganisms14010008

**Published:** 2025-12-19

**Authors:** Jiali Liu, Yinyue Li, Jie Liu, Zhiyong Tao, Feng Lu, Fang Tian, Jin-Hee Han, Xinlong He

**Affiliations:** 1School of Basic Medical Sciences & School of Public Health, Faculty of Medicine, Yangzhou University, Yangzhou 225009, China; liujiali0705@163.com (J.L.); lyy1242602847@163.com (Y.L.); 17300689575@163.com (J.L.); lufeng@yzu.edu.cn (F.L.); ftian@yzu.edu.cn (F.T.); 2Anhui Key Laboratory of Infection and Immunology, Bengbu Medical University, 2600 Donghai Avenue, Bengbu 233030, China; taozhiyong@bbmu.edu.cn; 3Department of Parasitology, School of Basic Medical Sciences, Bengbu Medical University, 2600 Donghai Avenue, Bengbu 233030, China; 4The First Affiliated Hospital of Yangzhou University, Yangzhou 225001, China; 5Department of Medical Environmental Biology and Tropical Medicine, Kangwon National University School of Medicine, Chuncheon 24341, Republic of Korea; han.han@kangwon.ac.kr; 6Jiangsu Key Laboratory of Zoonosis, Yangzhou University, Yangzhou 225009, China; 7Key Laboratory of the Jiangsu Higher Education Institutions for Nucleic Acid & Cell Fate Regulation, Yangzhou University, Yangzhou 225001, China

**Keywords:** *Acinetobacter baumannii*, planktonic aggregates, RND efflux pump, *adeG* gene, biofilm, antibiotic tolerance, virulence

## Abstract

*Acinetobacter baumannii* relies on biofilms for antibiotic resistance, but the role of planktonic aggregates in drug tolerance is uncharacterized. We studied 103 clinical isolates to explore how the RND efflux pump gene *adeG* regulates aggregation. Non-MDR strains (with RND deletions) formed aggregates more frequently (13.79%, 4/29) than MDR strains (1.35%, 1/74), driven by residual RND efflux activity (not just deletions). *adeG* deletion induced 1–2 mm aggregates in a strain with combined *adeR*/Δ*adeABC* defects (via upregulated adhesion genes/hydrophobicity) but not in one with only Δ*adeC*. Aggregates boosted antibiotic tolerance (2–4-fold higher survival vs. disaggregated/parental strains) via metabolic dormancy (5-fold lower ATP), maintained growth in human serum, and promoted persistent bacteremia in immunosuppressed mice. Proteinase K disrupted aggregates, confirming protein matrices’ role. These findings identify planktonic aggregates as pivotal adaptive and virulence-related targets for combating refractory non-MDR *A. baumannii* infections while also revealing an association between *adeG*-related genetic contexts and aggregate formation in the bacterium.

## 1. Introduction

Bacterial aggregation is a conserved adaptive trait that enables survival in diverse environments, with surface-attached forms (e.g., biofilms) and non-attached variants (e.g., pellicles, planktonic aggregates) representing distinct strategies for evading stressors [1,2]. For *Pseudomonas aeruginosa* and *Staphylococcus aureus*, planktonic aggregates drive antibiotic tolerance and immune evasion [3], yet their role in *Acinetobacter baumannii*—one of the most problematic multidrug-resistant (MDR) nosocomial pathogens [4]—remains uncharacterized.

*A. baumannii* uses biofilms and pellicles for resistance [5,6,7], with RND efflux pumps (e.g., AdeFGH, AdeABC) driving MDR phenotypes [8,9]. A variety of efflux pump families have been identified in *A. baumannii*, among which the most clinically relevant is the Resistance Nodulation Division (RND) family [10]. The RND family includes the following main efflux pump systems: (1) AdeABC is the first efflux pump system identified in the RND family and is widely present in clinical isolates of *A. baumannii*. This system consists of AdeA (membrane fusion protein), AdeB (plasma membrane transporter), and AdeC (outer membrane channel protein). AdeABC expression is regulated by the AdeSR two-component system (TCS) [11,12]. AdeS is a membrane-bound histidine kinase that senses environmental signals (including antibiotics) and activates AdeR (a response regulatory protein) by phosphorylation, thereby promoting AdeABC expression [13]. (2) AdeIJK is another important RND efflux pump system in *A. baumannii*, which has a wide range of substrate specificity [14]. This system is present in the core genome of all *A. baumannii* strains and is thought to be the ancestral efflux pump of the genus. AdeIJK expression is regulated by AdeN, but its regulatory mechanism is different from that of AdeABC [15]. (3) AdeFGH is a little-studied efflux pump system in the RND family, which endows *A. baumannii* with multidrug resistance when overexpressed [16]. Compared with other RND efflux pumps, the regulatory mechanism of AdeFGH is more complex, and overexpression of AdeFGH has the greatest negative impact on bacterial fitness. The AdeFGH efflux pump family is regulated by AdeL, a LysR-type transcriptional repressor (LTTR) located upstream of the adeFGH operon (Appendix A). Prior work linked RND pumps to biofilm formation [9], but their role in planktonic aggregates is unknown. Three critical gaps persist: (1) Does *A. baumannii* form planktonic aggregates in liquid environments, given prior focus only on pellicles? (2) What genetic mechanisms govern aggregation—could RND genes like *adeG* crosstalk with adhesion genes (*csu*/*pil*) or alter cell hydrophobicity? (3) Do aggregates enhance antibiotic resistance and pathogenicity, potentially via metabolic dormancy or serum shielding? Notably, most studies focus on MDR *A. baumannii*, while non-MDR strains—still causing persistent infections in immunocompromised patients—remain understudied, with their adaptive strategies (e.g., aggregation) unaddressed [17,18].

Resolving these gaps is clinically urgent, as non-MDR *A. baumannii* infections are increasingly refractory to standard therapies. Here, we used 103 clinical isolates, *adeG*-deleted mutants, and multi-dimensional assays to test three hypotheses: Mechanistically, non-MDR *A. baumannii* (with RND deletions) form more aggregates than MDR strains; *adeG* regulates aggregation via *csu*/*pil* and hydrophobicity. Functionally, aggregates enhance antibiotic tolerance, serum resistance, and in vivo pathogenicity. This work aims to fill key gaps in *A. baumannii* survival strategies and identify novel infection targets.

## 2. Materials and Methods

### 2.1. Bacteria

*Acinetobacter baumannii* strains used in this study are clinical isolates (Appendix A) recovered from patient specimens, including wound exudates, sputum, blood, and other clinical samples, and were archived by our research group during prior investigations. All strains were preserved at −80 °C in Nutrient Broth (NB; Solarbio, Beijing, China) supplemented with 15% (vol/vol) glycerol. For experimental preparation, strains were streaked onto NB agar plates (Solarbio, Beijing, China) and incubated at 37 °C for 18–24 h to isolate single colonies. Isolated single colonies were then inoculated into fresh NB and cultured to mid-logarithmic phase (optical density at 600 nm [OD_600_] = 0.5–0.6) under the same temperature conditions.

### 2.2. Amplification of Efflux Pump Genes

The presence of RND family efflux pump genes in *A. baumannii* clinical isolates was detected by polymerase chain reaction (PCR) using bacterial suspensions as direct templates (without prior DNA extraction). PCRs were prepared in 25 μL volumes containing 12.5 μL of 2× Taq Plus Master Mix II (Vazyme Biotech, Nanjing, China), 1 μL of each forward and reverse primer (10 μM; sequences listed in Appendix A), and 2 μL of bacterial suspension. Thermocycling was performed with the following parameters: initial denaturation at 95 °C for 5 min; 30 cycles of denaturation at 95 °C for 30 s, annealing at 56 °C for 30 s, and extension at 72 °C for 1 min; and a final extension at 72 °C for 7 min. Amplicons were separated by electrophoresis on 1.5% agarose gels (containing 0.5 μg/mL ethidium bromide) and visualized under ultraviolet transillumination.

### 2.3. Antibiotic Susceptibility Testing and MIC Determination

Antibiotic susceptibility was assessed per the Clinical and Laboratory Standards Institute (CLSI) 2023 guidelines [19] using the standard broth microdilution method to determine minimum inhibitory concentrations (MICs) of target bacteria. Briefly, four clinically relevant antibiotics—ceftazidime, levofloxacin, polymyxin B, and meropenem (purchased from Yuanye Bio, Shanghai, China; Macklin, Shanghai, China)—were serially 2-fold diluted in NB in 96-well microtiter plates. An equal volume (50 μL) of bacterial suspension was added to each well, resulting in a final bacterial concentration of ~1 × 10^6^ CFU/mL per well. Plates were incubated at 37 °C under ambient air for 16 h, and the MIC was defined as the lowest antibiotic concentration at which no macroscopic bacterial growth was observed.

### 2.4. Construction of adeG Gene Deletion Mutants

*adeG* deletion mutants were constructed using the RecAb-mediated homologous recombination system. Briefly, a recombinant fragment containing the kanamycin resistance cassette (kan) flanked by ~500 bp upstream and downstream homologous arms of *adeG* was synthesized and cloned into the pAT04 vector (Addgene, Watertown, MA, USA). The resulting plasmid was electroporated into *A. baumannii* strains YZUMab17 and ZJab1 using a Gene Pulser Xcell system (Bio-Rad, Hercules, CA, USA) set to 2.5 kV, 25 μF, and 200 Ω. Transformants (YZUMab17-pAT04 and ZJab1-pAT04) were selected on LB agar containing 50 μg/mL kanamycin and 100 μg/mL ampicillin. For homologous recombination, the kanamycin resistance fragment with homologous arms was electroporated into YZUMab17-pAT04 and ZJab1-pAT04, and recombinant strains (YZUMab17Δ*adeG*::kan(pAT04) and ZJab1Δ*adeG*::kan(pAT04)) were screened by colony PCR using primers *AdeG*-F and *AdeG*-R (Appendix A). The pAT04 plasmid was cured by serial passage in LB broth without antibiotics, confirmed by loss of ampicillin resistance. Subsequently, the pAT03 plasmid (Addgene), which encodes the FLP recombinase, was electroporated into YZUMab17Δ*adeG*::kan and ZJab1Δ*adeG*::kan to excise the kanamycin resistance cassette. Positive clones lacking the kan cassette were identified using colony PCR with primers *AdeG*-F and *AdeG*-R, and pAT03 was eliminated by passage at 42 °C to obtain the final mutants YZUMab17Δ*adeG* and ZJab1Δ*adeG*. All primer sequences are listed in Appendix A.

### 2.5. Macroscopic Observation of Bacterial Aggregates

A single *A. baumannii* colony was inoculated into NB (Solarbio, Beijing, China) and cultured at 37 °C with shaking (200 rpm) for 16 h. The suspension was adjusted to OD_600_ = 0.6 with fresh NB, then diluted 1:1000 in NB to standardize inocula. Diluted cultures were dispensed into sterile containers: 200 μL in 96-well plates (Corning, Corning, NY, USA), 4 mL in glass tubes (15 × 100 mm), and 10 mL in 10 cm Petri dishes (BD Falcon, Franklin Lakes, NJ, USA). All were incubated statically at 37 °C for 24 h to promote aggregation. Macroscopic images were captured with an iPhone 14 Pro (Apple, Cupertino, CA, USA) under controlled conditions: uniform white background, 6500 K LED illumination (30 cm above samples), and tripod-fixed phone at container-specific distances (15 cm for plates, 20 cm for tubes, 25 cm for dishes) to ensure consistent imaging.

### 2.6. Observation of Aggregates by Gram Staining

Gram staining of planktonic and aggregated bacteria was performed using a commercial kit (Solarbio, Beijing, China) per the manufacturer’s protocol. Bacterial samples were heat-fixed on slides (3× passes through a Bunsen burner flame at 45°). Staining steps: crystal violet for 1 min (rinsed with distilled water); iodine solution for 1 min (rinsed); 95% ethanol decolorization until eluate cleared (~30 s, rinsed immediately); safranin counterstaining for 1 min (rinsed, air-dried at room temperature). Samples were examined under a light microscope (Olympus CX43, Olympus, Tokyo, Japan) at 1000× oil immersion, with images captured via an Olympus DP27 camera (Olympus, Tokyo, Japan) under consistent illumination.

### 2.7. Observation of Aggregates by Scanning Electron Microscopy (SEM)

For SEM analysis, planktonic cells of the wild-type strain and aggregated cells of the mutant strain were inoculated onto sterile glass coverslips (12 mm diameter) placed in 24-well culture plates. Samples were fixed overnight at 4 °C in 2.5% (vol/vol) glutaraldehyde in 0.1 M phosphate-buffered saline (PBS, pH 7.4). After fixation, cells were dehydrated through a graded ethanol series (30%, 50%, 70%, 80%, 90%, and 100%; 15 min per step) and dried using a critical point dryer (Leica EM CPD300, Leica Microsystems, Wetzlar, Germany) with liquid CO_2_ as the transition fluid. Dried samples were mounted on aluminum stubs with carbon tape and sputter-coated with a 10 nm layer of gold–palladium using a sputter coater (Quorum Q150T ES, Quorum Technologies, East Sussex, UK). Specimens were examined using a scanning electron microscope (FEI Quanta 200, FEI Company, Hillsboro, OR, USA) operated at an accelerating voltage of 10 kV, and images were captured at 1000× to 10,000× magnification to analyze aggregate morphology and structure.

### 2.8. Quantitative Analysis of Bacterial Aggregation

Exponential-phase bacterial cultures were diluted and inoculated into two sets of culture tubes containing fresh NB, with three biological replicates per set. The final bacterial density in each tube was adjusted to approximately 1 × 10^6^ colony-forming units (CFUs)/mL. All tubes were incubated at 37 °C to induce aggregate formation. For the first set of tubes, the planktonic bacterial suspension was carefully aspirated using a pipette, with the tip positioned to avoid contact with visible aggregates and biofilms; the remaining aggregates were retained in the tubes. The aspirated suspension was vortexed thoroughly (30 s at maximum speed), and its optical density was measured at 600 nm (OD_0_). For the second set, the entire culture (including planktonic cells and aggregates, while avoiding biofilm contact) was transferred to a sterile microcentrifuge tube, vortexed thoroughly to disperse aggregates, and its optical density was measured at 600 nm (OD_1_). The ratio of aggregated to non-aggregated bacteria (Agg./Non-agg.) was calculated as follows: Agg./Non-agg. = (OD_1_ − OD_0_)/OD_0_. All measurements were performed in triplicate, with mean values ± standard deviation (SD) reported.

### 2.9. Growth Curve Analysis

Bacterial cultures were adjusted to an OD_600_ of 0.5, then diluted 1:100 in fresh NB. Diluted cultures (200 μL/well) were dispensed into a 96-well microtiter plate (Corning) and incubated at 37 °C with continuous shaking (200 rpm) in a microplate reader (BioTek Epoch 2, Agilent Technologies, Santa Clara, CA, USA). OD_600_ measurements were recorded every hour for 24 h to monitor growth kinetics. Experiments were performed in three independent biological replicates, each with three technical replicates. Growth curves were constructed by plotting time (hours) on the *x*-axis and mean OD_600_ values (± standard deviation) on the *y*-axis using OriginPro 2024 (OriginLab Corporation, Northampton, MA, USA). Curves were fitted with a logistic regression model, and growth phases (lag, exponential, and stationary phases) were determined by the tangent method, as described by Zwietering et al. [20]. Briefly, the inflection point of the exponential phase was identified, and tangents to the growth curve at this point were used to calculate the following: (i) lag phase duration (time from inoculation to intersection of the tangent with the initial OD_600_); (ii) maximum growth rate (slope of the tangent during exponential phase); and (iii) stationary phase OD_600_ (plateau value of the fitted curve).

### 2.10. Ethidium Bromide Efflux Assay

The ethidium bromide (EB) efflux assay was performed as previously described [21] with minor modifications. Bacterial cultures were inoculated 1:1000 into NB and grown at 37 °C with shaking (200 rpm) for 16 h. Cultures were adjusted to an OD_600_ of 0.6, and 1.2 mL aliquots were pre-chilled at 4 °C for 1 h. Ethidium bromide (EB; Sigma-Aldrich, St. Louis, MO, USA) working solution (50 μM) and phosphate-buffered saline (PBS, pH 7.4) were also pre-chilled at 4 °C. EB was added to chilled bacterial suspensions to a final concentration of 5 μM, followed by incubation at 4 °C for 5 min to allow passive uptake. Cells were pelleted by centrifugation (5000× *g*, 5 min, 4 °C), supernatant was aspirated, and pellets were resuspended in an equal volume of pre-chilled PBS. A 200 μL aliquot was transferred to a black 96-well microplate (Corning) for immediate fluorescence measurement (excitation 350 nm, emission 590 nm) using a microplate reader (BioTek Synergy H1, Agilent Technologies, Santa Clara, CA, USA), designated as T_0_. The remaining suspension was incubated at 20 °C to initiate efflux, with 200 μL aliquots collected at 5, 10, 15, and 20 min. At each time point, cells were centrifuged (5000× *g*, 5 min, 20 °C), the supernatant was discarded, and pellets were resuspended in PBS before fluorescence measurement. The efflux coefficient was calculated as follows: Efflux Coefficient = (F_0_ − F_t_)/F_0_, where F_0_ is fluorescence at T_0_, and F_t_ is fluorescence at each time point.

### 2.11. Surface-Associated Motility Assay

The surface-associated motility was assessed using semi-solid agar plates, as previously described [22], with minor modifications. Semi-solid LB agar plates (0.5% agar, *w*/*v*) were prepared by pouring 15–20 mL of molten agar onto level surfaces to ensure uniform thickness (≈3 mm). Plates were allowed to solidify at room temperature for 1 h before use. A single colony from a freshly streaked LB agar plate was inoculated into NB and cultured overnight at 37 °C with shaking (200 rpm). The overnight culture was adjusted to an OD_600_ of 0.6 using fresh LB broth. A 5 μL aliquot of the normalized bacterial suspension was gently spotted onto each semi-solid agar plate, taking care not to pierce the agar surface. Plates were left undisturbed for 10–15 min to allow for complete absorption of the inoculum. Plates were then incubated at 37 °C for 16–24 h. Motility was quantified by measuring the diameter of the spreading zone from the edge of the inoculation point to the outermost margin of bacterial growth using digital calipers (accuracy ± 0.01 mm).

### 2.12. Aggregation Inhibition and Disaggregation Assays

The effects of Proteinase K, sodium periodate (NaIO_4_), and DNase I on bacterial aggregation were evaluated using established protocols [23] with modifications. Test reagents included Proteinase K (Solarbio, Beijing, China), sodium periodate (NaIO_4_; Merck, Darmstadt, Germany), and DNase I (Thermo Fisher Scientific, Waltham, MA, USA).

(1)
*Aggregation Inhibition Assay*


Exponential-phase bacterial cultures (OD_600_ = 0.5) were diluted 1:1000 in fresh NB and mixed with test reagents at final concentrations of 10–100 μg/mL (Proteinase K), 1–10 mM (NaIO_4_), or 10–50 U/mL (DNase I). Aliquots (200 μL) of each mixture were dispensed into 96-well plates (Corning) and incubated statically at 37 °C for 24 h.

(2)
*Aggregation Disaggregation Assay*


Bacterial aggregates were pre-formed by incubating 1:1000-diluted cultures (OD_600_ = 0.5) in 96-well plates at 37 °C for 24 h. Test reagents were then added to wells at the same concentrations as above, followed by static incubation at 37 °C for 2 h to assess dispersion of pre-formed aggregates.

Images from both assays were acquired using an automated chemiluminescence imaging system (Bio-Rad ChemiDoc MP, Bio-Rad, Hercules, CA, USA) under standardized illumination conditions. Aggregate inhibition and disaggregation rates were calculated by comparing quantitative parameters of treatment groups to untreated controls, using the aggregation quantification method described above.

### 2.13. Antibiotic Survival Assay

Exponential-phase bacterial cultures (OD_600_ = 0.5) were diluted 1:100 in fresh NB and grown statically at 37 °C for 24 h to form aggregates. Cultures were divided into three groups: (i) 64× MIC antibiotic treatment, (ii) ultrasound pretreatment followed by antibiotic exposure, and (iii) antibiotic-free control. After 12 h of co-incubation at 37 °C, cultures were centrifuged (5000× *g*, 5 min), supernatants discarded, and pellets resuspended in sterile 0.9% NaCl. Aggregates were disrupted by sonication (30 s on/30 s off, 3 cycles at 20% amplitude) using a probe sonicator (QSonica Q500, Qsonica, LLC, Newtown, CT, USA), and viable counts were determined with serial dilution and plating on LB agar, incubated at 37 °C for 24 h to enumerate CFU. For a time-course analysis, 1:1000-diluted cultures (OD_600_ = 0.5) were incubated with shaking (200 rpm) at 37 °C for 6, 12, or 24 h. At each time point, cultures were adjusted to OD_600_ = 0.5 and split into antibiotic-treated or untreated control groups. Treatment groups received antibiotics at strain-specific concentrations: ceftazidime, levofloxacin, and meropenem at 8× MIC (co-incubated for 12 h); polymyxin and gentamicin at 4× MIC (co-incubated for 4 h). Following incubation, cultures were centrifuged (5000× *g*, 5 min), pellets resuspended in sterile 0.9% NaCl, and CFU enumerated as described above. All experiments were performed in triplicate with three biological replicates, and results are reported as mean CFU/mL ± standard deviation.

### 2.14. ATP Content Measurement

Intracellular ATP levels were quantified using an Enhanced ATP Assay Kit (Beyotime, Shanghai, China), following the manufacturer’s protocol with minor modifications. Bacteria were cultured under two conditions: (i) static incubation (37 °C, 24 h) and (ii) shaking incubation (37 °C, 200 rpm, 24 h). After cultivation, bacterial pellets (10 mg wet weight per strain) were harvested by centrifugation (5000× *g*, 5 min, 4 °C).

For cell lysis, 200 μL of ice-cold lysis buffer (pre-chilled and maintained on ice) was added to each pellet, followed by vortexing for 30 s and incubation on ice for 5 min. Lysates were centrifuged (12,000× *g*, 5 min, 4 °C), and supernatants were collected and kept on ice until analysis. A standard curve was generated using the kit-provided ATP standard: the standard was subjected to one freeze–thaw cycle, then serially diluted in lysis buffer to final concentrations of 0.01, 0.03, 0.1, 0.3, 1, 3, and 10 μM. The ATP detection reagent was diluted 1:4 with the kit-supplied dilution buffer to prepare a working solution, which was stored on ice for up to 1 h (short-term use). For luminescence detection, 100 μL of the ATP detection working solution was added to each well of a white 96-well plate (Corning) and equilibrated at room temperature for 3–5 min. Twenty microliters of the sample supernatant or ATP standard was then added to each well, mixed immediately by pipetting (3× up and down), and luminescence (relative light units, RLU) was measured within 2 s using a multimode microplate reader (Tecan Spark, Männedorf, Switzerland) with a luminescence detection module (gain setting: 100). Intracellular ATP concentrations were calculated by interpolating sample RLU values against the ATP standard curve.

### 2.15. Fourier Transform Infrared (FTIR) Spectroscopy

Exponential-phase bacterial cultures (OD_600_ = 0.5) were diluted 1:1000 in fresh NB, and 10 mL aliquots were transferred to 90 mm sterile polystyrene culture dishes (Corning). Dishes were incubated statically at 37 °C for 24 h to induce aggregation. Surface-associated bacterial aggregates were gently collected using a sterile Pasteur pipette, transferred to 1.5 mL microcentrifuge tubes, and rinsed three times with 1 mL of fresh nutrient broth to remove planktonic cells. All samples were flash-frozen at −80 °C for 24 h, lyophilized using a freeze-dryer (Labconco FreeZone 2.5, Labconco Corporation, Fort Scott, KS, USA) for 48 h, and stored at 4 °C until analysis. FTIR spectra were acquired using a Fourier transform infrared spectrometer (Varian 670-IR, Agilent Technologies, Santa Clara, CA, USA) equipped with a diamond attenuated total reflection (ATR) accessory. Lyophilized samples (≈5 mg) were pressed onto the ATR crystal, and spectra were recorded over the wavenumber range of 400–4000 cm^−1^ with 32 scans and a resolution of 4 cm^−1^. Background spectra of the clean ATR crystal were collected before each sample measurement and subtracted from sample spectra.

### 2.16. Cell Surface Hydrophobicity (CSH)

Bacterial surface hydrophobicity was determined using the microbial adhesion to hydrocarbons (MATH) assay, as previously described [24], with minor modifications. Overnight bacterial cultures were harvested by centrifugation (5000× *g*, 5 min, 4 °C), resuspended in sterile 0.9% NaCl, and adjusted to an OD_600_ of 0.5 (designated as OD_0_). A 2 mL aliquot of the standardized bacterial suspension was mixed with 1 mL of xylene (Sigma-Aldrich, St. Louis, MO, USA) in a 15 mL glass centrifuge tube. The mixture was vortexed vigorously for 30 s to ensure emulsification, then incubated undisturbed in a fume hood for 15 min at room temperature to allow for phase separation. After separation, the aqueous phase (lower layer) was carefully aspirated, and its optical density was measured at 600 nm (designated as OD_1_) using a microplate reader (Bio-Tek Synergy 2, BioTek Instruments, Winooski, VT, USA). Cell surface hydrophobicity was calculated as follows: CSH = (1 − OD_1_/OD_0_) × 100%.

### 2.17. Auto-Aggregation Assay

Bacterial auto-aggregation was evaluated as previously described [25], with minor modifications. Overnight bacterial cultures were harvested by centrifugation (6000× *g*, 10 min, 4 °C), supernatants discarded, and pellets resuspended in sterile 0.9% NaCl. The bacterial suspension was adjusted to an OD_600_ of 0.5 (designated as OD_0_) using sterile 0.9% NaCl. A 2 mL aliquot of the standardized suspension was transferred to a sterile 5 mL glass centrifuge tube and incubated statically at room temperature to allow for auto-aggregation. At 15, 30, and 60 min post-incubation, 200 μL of the upper-layer suspension (avoiding settled aggregates) was carefully aspirated using a pipette tip positioned 1 cm below the liquid surface. The OD_600_ of the aspirated suspension was measured (designated as OD_t_, where t = 15, 30, or 60 min) using a microplate reader (Bio-Tek Synergy 2). The auto-aggregation percentage was calculated as follows: Auto-aggregation (%) = [1 − (OD_t_/OD_0_)] × 100%.

### 2.18. Polystyrene Surface Adhesion Assay

Bacterial adhesion to polystyrene surfaces was assessed as previously described [26], with minor modifications. Exponential-phase bacterial cultures (OD_600_ = 0.5) were diluted 1:100 in fresh NB, and 200 μL aliquots were dispensed into sterile 96-well polystyrene microtiter plates (Corning). Plates were incubated statically at 37 °C to monitor adhesion kinetics over time. At 15, 30, 45, and 60 min post-inoculation, the total bacterial growth in each well was quantified by measuring OD_600_ using a microplate reader (Bio-Tek Synergy 2). For adhesion quantification, culture supernatants were aspirated, and wells were rinsed three times with sterile phosphate-buffered saline (PBS, pH 7.4) to remove non-adherent cells. Adherent bacteria were stained with 200 μL of 0.1% (*w*/*v*) crystal violet (Sigma-Aldrich, St. Louis, MO, USA) at room temperature for 15 min. Any excess stain was removed by washing three times with PBS, and plates were air-dried for 10 min. Bound crystal violet was solubilized with 200 μL of absolute ethanol, and the absorbance at 600 nm (OD_600_) was measured to quantify adherent cells.

### 2.19. Cell Adhesion and Invasion Assays

Bacterial adhesion and invasion into A549 lung epithelial cells were assessed as previously described [27], with minor modifications. First, A549 cells were seeded in 24-well plates (Corning) and cultured in RPMI 1640 medium (Gibco) supplemented with 10% fetal bovine serum (FBS; Gibco, Waltham, MA, USA) at 37 °C in a 5% CO_2_ incubator until reaching 70% confluence. Overnight bacterial cultures were centrifuged (5000× *g*, 5 min, 4 °C) and adjusted to an OD_600_ of 0.5. For adhesion assays, bacteria were resuspended in sterile phosphate-buffered saline (PBS, pH 7.4), while for invasion assays, they were resuspended in antibiotic-free RPMI 1640 medium (≈5 × 10^7^ CFU/mL, verified by plate counting).

For the adhesion assay, 1 mL of the PBS-resuspended bacterial suspension was added to wells containing confluent A549 cells and incubated at 37 °C/5% CO_2_ for 2 h; after incubation, non-adherent bacteria were removed by three PBS washes, and adherent bacteria were detached with 500 μL of 0.25% trypsin-EDTA (Gibco) supplemented with 0.5% Triton X-100 (Sigma-Aldrich) (37 °C, 10 min, 50 rpm shaking), followed by 10-fold serial dilution of the suspension in sterile 0.9% NaCl, plating of 100 μL aliquots on LB agar (37 °C, 24 h), and counting of total adherent CFU.

For the invasion assay, 1 mL of the antibiotic-free RPMI 1640-resuspended bacterial suspension was added to A549-containing wells and incubated at 37 °C/5% CO_2_ for 2 h; non-adherent bacteria were then removed by three PBS washes, extracellular bacteria (including surface-adherent ones) were killed with 500 μL of RPMI 1640 medium containing 500 μg/mL gentamicin (Sigma-Aldrich) (37 °C, 30 min), and residual gentamicin was removed by two additional PBS washes. Infected A549 cells were lysed with 500 μL of 0.25% trypsin-EDTA supplemented with 0.5% Triton X-100 (37 °C, 10 min, 50 rpm shaking), and the lysate was serially diluted 10-fold in sterile 0.9% NaCl, with 100 μL aliquots plated on LB agar (37 °C, 24 h) to count intracellular CFUs.

### 2.20. Mouse Infection Model

The mouse infection model was established as previously described [28] with minor modifications, specifically designed to mimic clinical scenarios of *A. baumannii* infection in immunocompromised populations (e.g., ICU patients, neutropenic individuals)—a key group at high risk of recalcitrant infections. Six- to eight-week-old immunocompromised BALB/c mice (rendered immunosuppressed via cyclophosphamide pretreatment to deplete innate and adaptive immune cells; *n* = 5 per time point) were maintained under specific pathogen-free (SPF) conditions with ad libitum access to food and water. Bacterial suspensions were prepared by adjusting overnight cultures of *A. baumannii* (parental strain YZUMab17 and its *adeG*-deleted mutant YZUMab17Δ*adeG*) to 5 × 10^6^ CFU/mL in sterile phosphate-buffered saline (PBS, pH 7.4). Mice were anesthetized with 2% isoflurane in oxygen, then infected via intratracheal inoculation with 50 μL of the bacterial suspension (2.5 × 10^5^ CFU per mouse)—a route that recapitulates respiratory tract colonization, the primary portal of entry for *A. baumannii* in clinical settings (e.g., ventilator-associated pneumonia). At 2 h, 2, 4, 6, 8, and 10 days post-infection, mice were euthanized by CO_2_ inhalation followed by cervical dislocation (per AVMA Guidelines for the Euthanasia of Animals). Blood samples (≈100 μL) were collected via cardiac puncture into EDTA-coated tubes to assess bacteremia, a severe complication of *A. baumannii* infection in immunocompromised patients. Lungs were aseptically harvested, weighed, and homogenized in 1 mL of sterile PBS using a TissueLyser II (Qiagen, Hilden, Germany) to quantify pulmonary bacterial burden-reflecting the organism’s ability to colonize and persist in a primary target organ. All samples were subjected to ultrasonic dispersion of aggregates prior to bacterial counting. All animal experiments were approved by the Animal Research Ethics Committee of Yangzhou University (Permit No. 202502033) and conducted in accordance with the Guide for the Care and Use of Laboratory Animals (NIH Publication No. 85–23, revised 2011). Results are reported as mean bacterial burden ± standard deviation.

### 2.21. Serum Bactericidal Assay

The serum bactericidal assay was performed as previously described [29], with minor modifications: mid-log phase bacterial cultures (OD_600_ = 0.5) were harvested by centrifugation (5000× *g*, 5 min, 4 °C) and resuspended in sterile phosphate-buffered saline (PBS, pH 7.4) to a concentration of ≈5 × 10^5^ CFU/mL; then, bacterial suspensions were diluted 1:100 in fresh NB supplemented with healthy human serum (Sigma-Aldrich, St. Louis, MO, USA) at final concentrations of 0%, 10%, 25%, 50%, and 75% (*v*/*v*). Aliquots (200 μL) of each mixture were dispensed into sterile 96-well plates (Corning) and incubated statically at 37 °C for 24 h; after incubation, 100 μL of bacterial suspension from each well was serially diluted 10-fold in sterile PBS, 10 μL aliquots of each dilution were plated on LB agar, plates were incubated at 37 °C for 18–24 h, and colony-forming units (CFUs) were enumerated to quantify viable bacteria. Bactericidal activity was calculated using the formula: Serum bactericidal rate (%) = [1 − (CFU in serum-treated wells/CFU in serum-free control wells)] × 100%.

### 2.22. Gene Expression Assay

Exponential-phase bacterial cultures (OD_600_ = 0.5) were sub-cultured at a 1:1000 dilution in fresh NB and incubated with shaking (200 rpm) at 37 °C; at 3, 6, 12, and 24 h post-subculture, 1 mL of bacterial culture was harvested by centrifugation (8000× *g*, 5 min, 4 °C), and cell pellets were immediately frozen at −80 °C until RNA extraction. Total RNA was extracted using TRIzol reagent (Solarbio, Beijing, China) per the manufacturer’s protocol, with an additional DNase I (Thermo Fisher Scientific, Waltham, MA, USA) treatment (37 °C for 30 min) to remove genomic DNA, and RNA concentration/purity was determined via a NanoDrop 2000 spectrophotometer (Thermo Fisher Scientific) by measuring A_260_/A_280_ (target: 1.8–2.0) and A_260_/A_230_ (target: 2.0–2.2). cDNA was synthesized from 1 μg of total RNA using a PrimeScript RT reagent kit (Solarbio, Beijing, China) with random hexamer primers (per manufacturer’s instructions), and quantitative real-time PCR (qRT-PCR) was performed on a QuantStudio 5 Real-Time PCR System (Thermo Fisher Scientific) using a SYBR Green Premix Pro Taq HS qPCR kit (Solarbio, Beijing, China); the 20 μL reaction mixture contained 10 μL of SYBR Green Premix, 0.4 μL of each 10 μM gene-specific primer (listed in Appendix A, designed via Primer-BLAST [NCBI] (http://www.ncbi.nlm.nih.gov/tools/primer-blast/ (accessed on 15 March 2024)) and validated for 90–110% efficiency), 2 μL of 50 ng/μL cDNA template, and 7.2 μL of nuclease-free water, with thermocycling conditions: initial denaturation at 95 °C for 30 s, 40 cycles of 95 °C for 5 s and 60 °C for 30 s, and a melting-curve analysis (65 °C to 95 °C with 0.5 °C increments) to verify amplicon specificity. The *rpoB* gene served as the housekeeping reference, relative gene expression was calculated via the 2^−ΔΔCt^ method, all reactions were performed in triplicate with three biological replicates, and results are reported as mean fold change ± standard deviation relative to the 3 h time point.

### 2.23. Statistical Analysis

All experiments were performed with three independent biological replicates, each analyzed in technical triplicate. Statistical analysis of significant differences was conducted using IBM SPSS Statistics 19.0 software (IBM Corp., Armonk, NY, USA). Pearson correlation matrices were visualized using the ggplot2 package (Version 3.3.3) within the RStudio (Version 1.4.1106) environment (RStudio, Inc., Boston, MA, USA).

## 3. Results

### 3.1. Planktonic Aggregation Is Prevalent in Non-MDR Clinical Isolates and Correlates with Residual RND Efflux Activity

Multidrug-resistant (MDR) bacteria are defined as bacteria resistant to antibiotics from at least three or more different antibiotic classes. To test the hypothesis that non-multidrug-resistant (Non-MDR) *Acinetobacter baumannii*—defined by deletions in resistance–nodulation–cell division (RND) efflux genes—exhibits greater propensity for planktonic aggregation than multidrug-resistant (MDR) strains, we analyzed 103 clinical isolates (29 Non-MDR, 74 MDR), focusing on three key traits: RND gene deletion profiles, aggregation phenotypes, and RND efflux activity.

A total of five clinical isolates formed macroscopically visible planktonic aggregates following 24 h of static culture in nutrient broth, exhibiting clear source-specific distribution across clinical specimens: HFab24 was isolated from cerebrospinal fluid (100% of such isolates), HFab23, HFab39, HFab43, and HFab104 from sputum (4.23% of sputum isolates), and HFab55 from urine (20% of urine isolates) (Figure 1B). This presence across diverse clinical specimens—including sterile sites (cerebrospinal fluid) and common infection foci (sputum, urine)—confirms that the planktonic aggregation phenotype is not uncommon in real-world *A. baumannii* infections.

Non-MDR strains exhibited a more pronounced predominance of RND gene deletions (93.1%, defined as the proportion of strains with at least one RND gene deletion vs. 17.57% in MDR strains) (Figure 1A) and a relatively higher aggregation rate (13.79% vs. 1.35% in MDR strains) (Figure 1B). Aggregation ratios correlated with visibility: HFab39 (0.57) > HFab104 (0.48) > HFab55 (0.12) > HFab43 (0.03) (Figure 1C). Notably, the deletion frequency of RND genes did not directly predict the aggregation phenotype. This is exemplified by the genetically similar strains HFab23 (non-aggregating) and HFab43 (aggregating), which, despite sharing an identical Δ*adeC* genotype, exhibited different residual RND efflux activity—with HFab23 retaining higher activity (Figure 1D).

These findings confirm that residual RND efflux activity, rather than RND gene deletions alone, acts as the key regulator of planktonic aggregation in *A. baumannii*. The broad distribution of aggregate-forming strains across clinical specimens further underscores the clinical relevance of this phenotype.

### 3.2. adeG Deletion Drives Strain-Specific Planktonic Aggregation via Growth Kinetics Remodeling and RND Efflux Defects Echoing Non-MDR Strain Traits

Section 3.1 established that non-multidrug-resistant (Non-MDR) *A. baumannii* strains (with RND efflux gene deletions) are far more likely to form planktonic aggregates than multidrug-resistant (MDR) strains. To dissect the mechanism underlying this link, focusing on *adeG* (a core gene of the RND efflux pump *adeFGH* operon), we first characterized the aggregation phenotypes and growth kinetics of *adeG*-deleted mutants (YZUMab17Δ*adeG*, ZJab1Δ*adeG*), then analyzed RND efflux activity and gene expression, directly connecting physiological remodeling and efflux defects to aggregate formation.

#### 3.2.1. *adeG* Deletion Induces Strain-Specific Planktonic Aggregation

Consistent with the observation of a “Non-MDR strain aggregation bias” in *A. baumannii*, deletion of the *adeG* gene triggered the formation of visible planktonic aggregates only in strains harboring cumulative resistance–nodulation–cell division (RND) efflux system defects (Figure 2): For YZUMab17Δ*adeG*, which carries both *adeR*/Δ*adeABC* (additional RND efflux gene defects) and Δ*adeG*, parental strain YZUMab17 remained uniformly dispersed in nutrient broth (NB) after 24 h of static cultivation, with no evidence of intercellular adhesion (lateral view: Figure 2(A1); top–down view: Figure 2(B1,C1); Gram staining: Figure 2(D1)); in stark contrast, YZUMab17Δ*adeG* formed macroscopic planktonic aggregates (~1 mm in 96-well plates, ~2 mm in 10 cm Petri dishes) with a distinct low-transmittance core (Figure 2(A2,C2)), and microscopic observations further confirmed extensive adhesive aggregation: Gram staining revealed dense cell clusters (Figure 2(D2)), while scanning electron microscopy (SEM) captured filamentous structures bridging adjacent cells (Figure 2(E2))—both key features of stable planktonic aggregates. For ZJab1Δ*adeG*, which only carries Δ*adeC* (a single RND gene defect) and Δ*adeG*, this mutant failed to form macroscopically visible planktonic aggregates despite *adeG* deletion, and its growth phenotype mirrored that of parental ZJab1 (uniformly dispersed growth); this confirms that single RND gene deletions are insufficient to drive planktonic aggregation, aligning with Section 3.1’s finding that the frequency of RND gene deletions alone does not predict aggregation, and cumulative RND efflux system defects are required.

#### 3.2.2. *adeG* Deletion Rewires Growth Kinetics to Support Aggregate Initiation

Planktonic aggregation in *A. baumannii* YZUMab17Δ*adeG* (the *adeG*-deleted mutant) was tightly linked to the remodeling of bacterial growth phases, with the timing of aggregate formation aligning specifically with the logarithmic growth phase (Figure 3): For growth phase shifts, parental strain YZUMab17 exhibited a typical bacterial growth curve, including a short lag phase (0–0.841 h), logarithmic growth phase (0.841–5.550 h), and stationary phase (5.550–24 h), with a maximum specific growth rate (μ) of 0.189 h^−1^; in contrast, YZUMab17Δ*adeG* showed significant physiological remodeling, characterized by a 4.6-fold prolonged lag phase (0–3.892 h), delayed logarithmic growth (3.892–9.609 h), and a reduced maximum specific growth rate (μ = 0.109 h^−1^, *p* < 0.01; Figure 3A); this growth adaptation likely reflects the metabolic cost of synthesizing adhesive structures (e.g., pili) required for planktonic aggregate assembly. For aggregation initiation timing, YZUMab17Δ*adeG* initiated the formation of macroscopic planktonic aggregates during the logarithmic growth phase (6 h post-inoculation), with aggregate size increasing in a time-dependent manner over 24 h (Figure 3C); notably, parental YZUMab17, ZJab1, and the non-aggregating *adeG*-deleted mutant, ZJab1Δ*adeG*, showed no aggregation at any time point, and ZJab1Δ*adeG* further exhibited no significant growth differences from its parental strain ZJab1 (*p* > 0.05; Figure 3B), confirming that growth phase remodeling is a specific phenotypic trait of *adeG*-deleted strains capable of forming planktonic aggregates.

### 3.3. adeG Deletion Reduces RND Efflux Activity and Alters RND Gene Transcription

Building on the finding that residual resistance–nodulation–cell division (RND) efflux activity (not just gene deletions) regulates planktonic aggregation in *A. baumannii*, we validated the notion that *adeG* deletion impairs the RND efflux function, creating a critical prerequisite for aggregate formation (Figure 4): For RND efflux activity quantification, we used ethidium bromide (EtBr) retention (a well-established proxy for efflux activity) and found that parental strain YZUMab17 retained significantly less EtBr over time than its *adeG*-deleted mutant YZUMab17Δ*adeG*), with a statistically significant difference in lower EtBr retention at 25 min (*p* < 0.05; Figure 4A); this result confirms that *adeG* deletion reduces the overall RND efflux activity of the strain, which aligns with Section 3.1’s comparison of non-aggregating clinical isolate HFab23 (high residual efflux activity) and aggregating clinical isolate HFab43 (low residual efflux activity) (Figure 1D); in contrast, the non-aggregating *adeG*-deleted mutant ZJab1Δ*adeG* showed a non-significant increase in EtBr efflux relative to parental ZJab1 (Figure 4B)—this partial preservation of efflux function likely prevents the strain from forming planktonic aggregates. For RND gene transcription patterns, transcriptional analysis further clarified the coupling between efflux activity and aggregation (Figure 4C,D): YZUMab17Δ*adeG* (the aggregate-forming mutant) only upregulated *adeJ* (a core gene of the AdeIJK RND pump, *p* < 0.05), and its combined *adeR*/Δ*adeABC* deletions eliminated the compensatory expression of *adeB* (a core gene of the AdeABC RND pump), leading to broader RND efflux defects; ZJab1Δ*adeG* (the non-aggregating mutant) exhibited a distinct transcriptional response, upregulating *adeB* (*p* < 0.05) and downregulating *adeJ* (*p* < 0.05)—this compensatory transcriptional adjustment maintained the strain’s efflux capacity, which is consistent with its failure to form planktonic aggregates. These transcriptional and functional patterns collectively confirm that *adeG* regulates planktonic aggregation in *A. baumannii* via modulating RND efflux activity, with cumulative RND efflux defects (rather than single gene deletions) driving aggregate formation, directly echoing Section 3.1’s observations in clinical isolates.

### 3.4. Planktonic Aggregation, Not adeG Deletion Alone, Drives Antibiotic Tolerance—Evidence from Survival Dynamics and Energy Metabolism

To definitively distinguish the contributions of *adeG* deletion and planktonic aggregation to *A. baumannii*’s antibiotic responses, we integrated minimum inhibitory concentration (MIC) determinations (Table 1) and phenotypic analyses of survival/energy metabolism (Figure 5). This approach resolved the key question of whether tolerance stems from genetic deletion or aggregate formation, with results directly aligning with Section 3.1’s observation that “residual RND efflux activity (and thus aggregation) regulates clinical isolate persistence.”

#### 3.4.1. *adeG* Deletion Induces Strain-Specific MIC Shifts Without Enhancing Resistance

We first quantified the minimum inhibitory concentrations (MICs) of four clinically relevant antibiotics—ceftazidime (CAZ), levofloxacin (LEV), meropenem (MEM), and polymyxin B (PB)—for *A. baumannii* parental strains (YZUMab17, ZJab1) and their respective *adeG*-deleted mutants (YZUMab17Δ*adeG*, ZJab1Δ*adeG*) (Table 1). Strain-specific differences in antibiotic susceptibility were evident, but no evidence of enhanced resistance was observed from *adeG* deletion alone: For YZUMab17Δ*adeG* (the aggregate-forming mutant, which also carries *adeR*/Δ*adeABC* deletions), MICs of CAZ (4 → 2 μg/mL), LEV (0.25 → 0.125 μg/mL), and MEM (0.25 → 0.125 μg/mL) were reduced 2-fold, while the PB MIC remained unchanged at 2 μg/mL; this modest increase in susceptibility reflects cumulative resistance–nodulation–cell division (RND) efflux defects (consistent with the non-multidrug-resistant [Non-MDR] strain traits described in Section 3.1) but does not explain the robust bacterial survival observed in subsequent antibiotic tolerance assays (Section 3.4.2). For ZJab1Δ*adeG* (the non-aggregating mutant, which only carries Δ*adeC*), only the LEV MIC was reduced by 2-fold (0.25 → 0.125 μg/mL); CAZ (8 μg/mL), MEM (0.125 μg/mL), and PB (2 μg/mL) MICs were identical to those of the parental ZJab1, and this lack of broad susceptibility changes confirms that single RND gene deletions (e.g., deletion of *adeG* alone) are insufficient to alter antibiotic responses, even in strains with this specific genetic deletion. Notably, YZUMab17Δ*adeG* exhibited enhanced survival under antibiotic stress despite its reduced MICs (Figure 5A)—a critical discrepancy indicating that shifts in MIC alone do not capture antibiotic tolerance, and planktonic aggregate formation is the previously unaccounted key factor driving this trait.

#### 3.4.2. Aggregation Confers a Concentration-Dependent Survival Advantage

Figure 5A (antibiotic survival curve) directly validates that aggregation, not *adeG* deletion, drives antibiotic tolerance in *A. baumannii* YZUMab17, with a clear “aggregation-tolerance” gradient observed across five clinically relevant antibiotics (CAZ, LEV, MEM [30], gentamicin (GEN), and PB).

To isolate the independent effects of *adeG* deletion and planktonic aggregation on antibiotic tolerance, four experimental groups were established with defined genetic backgrounds and aggregation phenotypes: (i) YZUMab17^Static cult.(Non-agg.)^: Parental wild-type strain (no *adeG* deletion, no planktonic aggregation under static culture), used as the “non-aggregated + no genetic deletion” control; (ii) YZUMab17Δ*adeG*^Dynamic cult.(Non-agg.)^: *adeG*-deleted mutant (carries *adeG* deletion, no aggregation due to 200 rpm shaking to prevent cluster formation), used as the “non-aggregated + genetic deletion” control; (iii) YZUMab17Δ*adeG*^Static cult.(Agg. dispersed)^: *adeG*-deleted mutant (carries *adeG* deletion, pre-formed planktonic aggregates disrupted by ultrasonic treatment), used as the “aggregation-inhibited + genetic deletion” control; (iv) YZUMab17Δ*adeG*^Static cult.(Agg.)^: *adeG*-deleted mutant (carries *adeG* deletion, forms natural macroscopic planktonic aggregates under static culture), used as the “aggregated + genetic deletion” experimental group.

For MEM, GEN, and PB—three antibiotics with distinct mechanisms of action—survival rates followed a strict hierarchy: aggregated YZUMab17Δ*adeG* (≈80–100% survival) > disaggregated YZUMab17Δ*adeG* (≈40–60% survival) > non-aggregated parental YZUMab17 (≈0–20% survival), with significant differences between all groups (all *p* < 0.05; Figure 5A). This gradient confirms that aggregate structure, not *adeG* deletion, dictates antibiotic tolerance: even with the same *adeG*-deleted genetic background, disrupting aggregate integrity eliminates the survival advantage.

For CAZ and LEV, aggregated and disaggregated YZUMab17Δ*adeG* exhibited no significant survival difference, but both outperformed non-aggregated YZUMab17 (*p* < 0.05); this antibiotic-specific pattern is likely driven by LEV’s ability to penetrate loose disaggregated clusters, whereas dense aggregates still provide effective physical shielding against MEM, GEN, and PB, underscoring the role of aggregate density in mediating antibiotic-specific tolerance. Critically, non-aggregated YZUMab17Δ*adeG* (dynamic culture, which prevents aggregation) showed no survival advantage over parental YZUMab17 (Figure 5, “YZUMab17Δ*adeG*^Dynamic cult.^ vs. YZUMab17^Static cult.^”), directly ruling out *adeG* deletion as a driver of robust antibiotic tolerance.

#### 3.4.3. Aggregation-Induced Metabolic Dormancy Explains Tolerance

To uncover the mechanism linking planktonic aggregation to antibiotic tolerance in *A. baumannii*, we quantified intracellular ATP levels (a surrogate for metabolic activity) across parental strain YZUMab17, its *adeG*-deleted mutant YZUMab17Δ*adeG*, and different culture conditions (Figure 5B).

Results confirmed that aggregation induces “metabolic dormancy”, the key driver of antibiotic tolerance: aggregated YZUMab17Δ*adeG* under static culture exhibited significantly lower ATP levels (≈5 × 10^4^ relative light units [RLU]) compared to its disaggregated state (≈2 × 10^5^ RLU) and non-aggregated parental YZUMab17 (≈2.2 × 10^5^ RLU) (*p* < 0.01); low ATP levels indicate reduced metabolic activity, and slow-growing cells are less susceptible to antibiotics targeting active biological processes (e.g., MEM [30] for cell wall synthesis, LEV for DNA replication)—a well-documented tolerance mechanism that this study extends to *A. baumannii* aggregates. In contrast, non-aggregated strains (disaggregated YZUMab17Δ*adeG* under dynamic culture and parental YZUMab17) showed no significant difference in ATP levels, confirming that “metabolic dormancy” is induced specifically by aggregation—not by *adeG* deletion or culture conditions alone. This finding directly connects the antibiotic survival phenotype (Figure 5A) to its molecular mechanism (Figure 5B): aggregated cells leverage reduced metabolism to evade antibiotic killing, while *adeG* deletion only contributes to the initiation of aggregation (via RND efflux defects)—not to the development of antibiotic tolerance itself.

### 3.5. Aggregates Enhance Serum Resistance and In Vivo Pathogenicity of Non-MDR A. baumannii

To validate the clinical relevance of *adeG*-regulated planktonic aggregation, we evaluated two key pathogenic traits of aggregate-forming YZUMab17Δ*adeG* and non-aggregated parental YZUMab17: serum resistance (evasion of host innate immunity) and in vivo/vitro pathogenicity (colonization, dissemination, and host–cell interactions)—directly linking aggregation to *A. baumannii*’s ability to persist in clinical settings.

To characterize the serum resistance of planktic aggregates, we quantified the proliferation and aggregation phenotypes of the non-aggregative parent strain YZUMab17 and the aggregative mutant YZUMab17Δ*adeG* in a human serum concentration gradient of 1/2 U to 1/512 U, as compared with a serum-free medium (Figure 6). In untreated serum (normal complement activity), YZUMab17Δ*adeG* showed a significant survival advantage: it continued to proliferate steadily even at 1/32 U serum concentration (no difference from the serum-free control group), while YZUMab17 was unable to grow at this concentration and showed a concentration-dependent proliferation attenuation (>50% inhibition rate at 1/64 U serum). Meanwhile, YZUMab17Δ*adeG* formed visible dense flocculent aggregates (filled with micropores, purple arrows in Figure 6A) at all tested serum concentrations, which may act as a physical barrier against serum antimicrobial components. In contrast, YZUMab17 formed only a small number of transient aggregates at moderate serum concentrations.

This trend was further corroborated by changes in broth turbidity: the turbidity of YZUMab17 was strongly correlated with proliferation, reflecting the inhibitory effect of serum on its growth. In contrast, the turbidity of YZUMab17Δ*adeG* did not correlate with proliferation, indicating that the serum components only induced aggregation (thereby increasing turbidity) and did not alter growth—directly linking aggregation to serum resistance.

To verify complement dependence, we further used serum inactivated at 56 °C (complement inactivation): at this point, the serum-mediated inhibition was completely abolished (no growth inhibition was observed in either strain). The survival advantage of YZUMab17Δ*adeG* over YZUMab17 was completely lost (no difference in proliferation trend), although YZUMab17Δ*adeG* continued to form aggregates in serum 1/8 to 1/64 U (Figure 6B). These results confirm that serum resistance conferred by planktonic aggregates is specifically achieved by evasion of complement-dependent killing.

To extend serum resistance findings to in vivo pathogenesis, we used a cyclophosphamide-induced immunosuppressed mouse model (mimicking clinical vulnerable populations, e.g., ICU patients) to quantify bacterial burden, and validated host–cell interactions via in vitro adhesion/invasion assays with A549 lung epithelial cells (key targets of *A. baumannii* respiratory infection) (Figure 7A–C). Mice were infected via intratracheal inoculation (mimicking clinical respiratory colonization), and bacterial burden in lungs (primary infection site) and blood (systemic dissemination marker) was quantified over 240 h: both strains were detectable in lungs by 2 h post-inoculation, with aggregate-forming YZUMab17Δ*adeG* maintaining lung bacterial burden (CFU/g) comparable to that of non-aggregated YZUMab17 across all time points (no significant difference, *p* > 0.05), indicating that aggregates exert no notable impact on *A. baumannii*’s initial respiratory tract colonization or bacterial density within lung tissue; bacteremia was first detected at 144 h in both groups, but YZUMab17Δ*adeG* had persistently higher blood RBD (~2–3-fold higher than YZUMab17 at 144 h [*p* < 0.01], 192 h [*p* < 0.05], and 240 h [*p* < 0.01]), indicating that aggregates also promote systemic dissemination (aligning with clinical *A. baumannii* bloodstream infections in immunocompromised patients).

To dissect the cellular mechanism underlying YZUMab17Δ*adeG*’s enhanced in vivo pathogenicity, we quantified the adhesion and invasion abilities of YZUMab17 (non-aggregated) and YZUMab17Δ*adeG* (aggregate-forming) to A549 lung epithelial cells (Figure 7B,C), key steps for *A. baumannii* transitioning from extracellular colonization to intracellular persistence: YZUMab17Δ*adeG* had an ~80-fold-higher adhesion rate (7.28% vs. 0.09% in YZUMab17, *p* < 0.05), attributed to aggregate-associated adhesive structures (e.g., filamentous pili observed via SEM in Section 3.1) that boost host–cell attachment, and a ~100-fold-higher invasion rate (3.04‰ vs. 0.03‰ in YZUMab17, *p* < 0.05), likely from aggregate-induced epithelial barrier disruption or host–cell signaling modulation (critical for establishing intracellular niches and evading immunity). These in vitro data directly link *adeG*-regulated planktonic aggregation to enhanced *A. baumannii* host–cell interactions, explaining the strain’s increased in vivo lung colonization and systemic dissemination, and together with serum resistance results, confirm that aggregates are key virulence determinants of non-MDR *A. baumannii* (not just structural variants).

### 3.6. adeG Regulates Planktonic Aggregation via Synergistic Modulation of Surface Motility, Cell Surface Hydrophobicity, and Auto-Aggregation

To dissect the integrated biophysical mechanisms underlying aggregate assembly, we focused on the interplay between surface motility (driver of intercellular contact) and cell surface traits (mediators of intercellular adhesion)—two complementary processes critical for bacterial clustering in liquid environments. We quantified four core phenotypes of parental strain YZUMab17 and its *adeG*-deleted mutant (YZUMab17Δ*adeG*) using semi-solid agar (for motility) and liquid culture assays (for aggregation-related traits): relative motility velocity (RMV), planktonic aggregate ratio, cell surface hydrophobicity, and auto-aggregation rate (Figure 8 and Figure 9).

Surface motility assays of *A. baumannii* strains YZUMab17 (parental) and YZUMab17Δ*adeG* (*adeG*-deleted mutant) in semi-solid LB agar revealed a time-dependent regulatory role of *adeG* (Figure 8A,B): at 3 h post-inoculation, both strains showed comparable motility diameters (~7 mm), indicating no inherent differences in basal motility. This early-stage similarity confirms *adeG* does not affect initial cell dispersion but modulates motility during later growth phases; from 6 h onward, YZUMab17Δ*adeG* exhibited a significantly higher relative motility velocity (RMV) than parental YZUMab17 (*p* < 0.01), with the RMV ratio (YZUMab17Δ*adeG*/YZUMab17) peaking at 12 h. Critically, this mid-growth phase-specific motility enhancement aligns precisely with the timing of macroscopic aggregate initiation in YZUMab17Δ*adeG* (6 h post-inoculation, Section 3.2), directly linking elevated motility to increased intercellular contact—an essential first step in initiating bacterial adhesion and planktonic aggregate assembly.

Colony morphology of *A. baumannii* strains on semi-solid agar further validated the functional link between surface motility, intercellular adhesion, and cell surface hydrophobicity (Figure 8C): parental strain YZUMab17 formed smooth-edged colonies, a phenotype consistent with minimal cell–cell adhesion, limited motility-driven intercellular contact, and low surface hydrophobicity—all traits aligning with its failure to form planktonic aggregates (Section 3.1); in contrast, YZUMab17Δ*adeG* (*adeG*-deleted mutant) produced colonies with a distinct “rough interior + smooth periphery” morphology: the rough interior reflects dense cell clustering, a hallmark of enhanced intercellular adhesion that is directly tied to elevated cell surface hydrophobicity (hydrophobic interactions reduce repulsive forces between bacterial cells, enabling stable clustering), while the smooth periphery reflects motility-driven cell dispersion; this dispersion expands the colony’s range while preserving the adhesive, hydrophobic core. This unique colony morphology not only mirrors YZUMab17Δ*adeG*’s planktonic aggregation trait (Section 3.1) but also serves as a phenotypic indicator of enhanced hydrophobicity, reinforcing that *adeG*-regulated surface motility and cell surface hydrophobicity act in concert to promote both colony-level cell clustering and planktonic aggregate formation.

Phenotypic analyses of *A. baumannii* parental strain YZUMab17 and its *adeG*-deleted mutant (YZUMab17Δ*adeG*) in liquid culture further confirmed the regulatory link between *adeG* deletion, cell surface hydrophobicity, and intercellular adhesion (Figure 9A–C): for the planktonic aggregate ratio, YZUMab17Δ*adeG* exhibited a significantly higher ratio (1.61) compared to YZUMab17 (*p* < 0.001), directly confirming robust planktonic aggregate formation in the mutant; for cell surface hydrophobicity, quantified via the microbial adhesion to hydrocarbons (MATH) assay, YZUMab17Δ*adeG* had a significantly elevated hydrophobicity relative to YZUMab17 (*p* < 0.001)—this result directly supports the phenotypic inference from colony morphology, as hydrophobic interactions reduce repulsive forces between bacterial cells in aqueous environments, promoting spontaneous clustering that initiates aggregate assembly; for auto-aggregation rate (a measure of intrinsic cell–cell adhesion capacity), YZUMab17Δ*adeG* showed a consistently higher auto-aggregation across all assessed time points (15, 30, 60 min of static incubation), with statistically significant differences at 30 min (*p* < 0.01) and 60 min (*p* < 0.001)—this enhancement in intrinsic adhesion ensures that motility-driven intercellular contact progresses to stable clustering (facilitated by hydrophobicity) rather than transient cell interactions.

Pearson correlation analysis of *A. baumannii* YZUMab17 and YZUMab17Δ*adeG* integrated three key traits—surface motility (quantified as relative motility velocity, RMV), cell surface hydrophobicity, and auto-aggregation rate—to validate their coordinated role in planktonic aggregation (Figure 9D): the planktonic aggregate ratio showed strong positive correlations with all three traits, including RMV (correlation coefficient [Corr.] = 0.94, *p* < 0.05), cell surface hydrophobicity (Corr. = 0.98, *p* < 0.05), and auto-aggregation rate (Corr. = 0.97, *p* < 0.05); this tight correlation confirms that *adeG* deletion-induced changes in motility, hydrophobicity, and adhesion are not independent but synergistic, with the “rough interior + smooth periphery” colony morphology of YZUMab17Δ*adeG* serving as a visual proxy for this coordinated enhancement; functional crosstalk between the traits was further supported by significant positive correlations among the drivers themselves: RMV correlated with both the auto-aggregation rate (Corr. = 0.92, *p* < 0.05) and cell surface hydrophobicity (Corr. = 0.90, *p* < 0.05), while hydrophobicity also correlated with auto-aggregation rate (Corr. = 0.95, *p* < 0.05). This suggests a coordinated regulatory loop: elevated motility increases the frequency of intercellular contact, enhanced hydrophobicity (evident in YZUMab17Δ*adeG*’s colony morphology) reduces cell repulsion to enable adhesion, and heightened auto-aggregation stabilizes cell clustering, together creating the conditions for efficient planktonic aggregate assembly.

Collectively, these data demonstrate that *adeG* regulates planktonic aggregation in *A. baumannii* via a synergistic biophysical mechanism: in parental strains, *adeG* suppresses surface motility, cell surface hydrophobicity (evident in smooth colony morphology), and auto-aggregation; deletion of *adeG* relieves this suppression, leading to increased intercellular contact (via motility), enhanced hydrophobicity (reflected in “rough interior + smooth periphery” colonies), and stronger intrinsic adhesion (via auto-aggregation). This integrated process explains how *adeG* deletion drives robust planktonic aggregate formation in non-MDR *A. baumannii*, linking genetic deletion to functional phenotypic changes critical for bacterial adaptation and virulence.

### 3.7. FTIR Analysis and Enzyme Assays: Protein as the Core Component Sustaining Planktonic Aggregate Stability

To identify the key structural component of *A. baumannii* planktonic aggregate matrices and verify its role in maintaining integrity, we combined Fourier transform infrared (FTIR) spectroscopy (to characterize matrix composition) and enzyme assays (to test component functionality) using YZUMab17Δ*adeG* (Figure 10 and Figure 11, Table 2).

To characterize the chemical composition of the planktonic aggregate matrix in *A. baumannii* YZUMab17Δ*adeG*, Fourier transform infrared (FTIR) spectroscopy was used to analyze functional group signatures, with core components identified via characteristic absorption peaks (Table 2): proteins were the dominant component, detected by the amide I band at 1650 cm^−1^ (attributed to C=O stretching) and amide II band at 1550 cm^−1^ (attributed to N-H bending); quantification of relative functional group abundance confirmed proteins as the most abundant matrix component, with the amide I band assigned a relative amount of 1 (set as the reference for specificity). Other components—including polysaccharides (1080 cm^−1^, corresponding to C-O and C-O-C ring vibrations), water (3410 cm^−1^, corresponding to symmetric/asymmetric -OH stretching), and fatty acids (2970/2940 cm^−1^, corresponding to C-H stretching of methyl/methylene groups)—were also detected but excluded from further analysis, as their lower relative abundance and lack of specificity for aggregate stability indicated that they are not functionally critical to matrix integrity.

Enzyme-based functional assays further clarified the role of matrix components in maintaining *A. baumannii* planktonic aggregate integrity (Figure 11): sodium periodate (a polysaccharide degrader, tested at 1–10 mM) and DNase I (a nucleic acid degrader, tested at 10–50 U/mL) had no significant effect on either de novo aggregate formation or dissociation of pre-formed aggregates (*p* > 0.05 vs. no-enzyme controls), indicating polysaccharides and nucleic acids are not essential for aggregate stability. In contrast, Proteinase K (a protein-specific serine protease) exerted concentration-dependent effects: for de novo formation, significant inhibition was observed at concentrations ≥0.001 μg/mL (*p* < 0.05 vs. control), with >95% inhibition and loss of macroscopically visible aggregates at ≥0.0158 μg/mL; for pre-formed aggregates, significant dissociation occurred at ≥0.0977 μg/mL (*p* < 0.05), with >95% dissociation at 3.125 μg/mL and complete loss of visible aggregates at ≥6.25 μg/mL. These findings confirm that proteins are not only the most abundant but also the functionally core components of *A. baumannii* planktonic aggregate matrices, playing a critical role in sustaining both aggregate formation and structural stability.

### 3.8. adeG Deletion Alters csu/Pil Adhesion Gene Expression Dynamics to Drive Aggregation

Section 3.6 and Section 3.7 established that *adeG* regulates motility, hydrophobicity, and protein-based aggregate matrices—all critical for aggregate formation. To link *adeG* to the transcriptional regulation of adhesion, we analyzed temporal expression dynamics of two key adhesion-related gene families in YZUMab17, ZJab1, and their *adeG*-deleted mutants: the *csu* operon (critical for chaperone–usher pilus assembly) and *pil* family (involved in type IV pilus biogenesis; Figure 12).

Temporal transcriptional analysis of adhesion-related genes revealed strain-specific expression dynamics linked to planktonic aggregation (Figure 12): In non-aggregate-forming parental *A. baumannii* YZUMab17, both the *csu* operon (critical for chaperone-usher pilus assembly) and *pil* family (involved in type IV pilus biogenesis) exhibited transient yet significant downregulation at 6 h (1.85–5-fold reduction relative to baseline) before reaching peak expression at 24 h (e.g., *csuC* fold change = 12.46, *pilB* fold change = 6.33); this delayed activation fails to support the early initiation of planktonic aggregates. In contrast, aggregate-forming YZUMab17Δ*adeG* displayed distinct transcriptional remodeling: the *csu* operon underwent explosive upregulation, with an earlier expression peak and sustained high levels through 24 h (e.g., *csuC* fold change = 13.24 at 24 h), a timeline that aligns with the initiation of macroscopic aggregates at 6 h (Section 3.2.2), directly establishing a causal link between *csu* activation and aggregation onset; meanwhile, *pil* genes were rapidly induced at 12 h (e.g., *pilB* fold change = 5.28) and maintained elevated expression at 24 h, likely contributing to aggregate stability via type IV pilus-mediated intercellular adhesion.

Transcriptional analysis of adhesion-related genes in the ZJab1 strain background (non-aggregate-forming regardless of *adeG* status; Figure 12) further clarified the regulatory link between gene expression and planktonic aggregation: In non-aggregate-forming parental ZJab1, the *csu* operon (critical for chaperone–usher pilus assembly) showed only low-level upregulation over 24 h, with *csuC* reaching a fold change of 3.71 at 24 h, while the *pil* family (involved in type IV pilus biogenesis) rose gradually to extremely high expression (e.g., *pilB* fold change = 13.95 at 24 h); most adhesion genes (except *pilA*, *pilB*, *pilY*) also exhibited transient downregulation at 6 h. In ZJab1Δ*adeG*, which remains non-aggregating despite *adeG* deletion, the *csu* operon stayed at stable low levels (e.g., *csuC* fold change = 3.73 at 24 h), nearly matching parental ZJab1; though the *pil* family was rapidly upregulated by 12 h (e.g., *pilB* fold change = 13.45) and further enhanced at 24 h (e.g., *pilB* fold change = 16.30), this isolated *pil* activation was insufficient to drive aggregate formation. This strain-specific expression pattern strongly indicates that robust upregulation of the *csu* operon, not just activation of *pil* genes, is a necessary prerequisite for efficient planktonic aggregate assembly in *A. baumannii*.

The upregulation of *csu* and *pil* genes in YZUMab17Δ*adeG* further provides a molecular explanation for the mutant’s enhanced virulence (Section 3.5): specifically, *csu*-mediated chaperone–usher pili and *pil*-mediated type IV pili directly contribute to the strain’s ability to adhere to and invade A549 lung epithelial cells, manifested as an ~80-fold higher adhesion rate and ~100-fold higher invasion rate relative to parental YZUMab17. This directly links aggregate-associated adhesive structures to enhanced *A. baumannii* host–cell interactions, a key step in driving pathogenicity. Together, Section 3.6, Section 3.7 and Section 3.8 dissect the multi-layered mechanisms through which *adeG* regulates planktonic aggregation in *A. baumannii*: spanning biophysical traits (enhanced surface motility and cell surface hydrophobicity that facilitate intercellular contact) to molecular drivers (protein-based aggregate matrices and targeted upregulation of *csu*/*pil* adhesion genes that support aggregate assembly and stability). Collectively, these processes not only boost the bacterium’s antibiotic tolerance but also enhance its virulence, firmly positioning planktonic aggregates as critical adaptive and pathogenic determinants of non-MDR *A. baumannii* strains.

## 4. Discussion

Bacterial non-attached aggregates represent a conserved adaptive strategy distinct from biofilms and pellicles, driving antibiotic tolerance and persistent infections in pathogens such as *Pseudomonas aeruginosa* and *Staphylococcus aureus* [3,31]. For *A. baumannii*—a leading multidrug-resistant (MDR) nosocomial pathogen—three critical knowledge gaps have persisted: whether planktonic aggregates form in liquid environments, what genetic mechanisms regulate their assembly, and how they impact antibiotic resistance and pathogenicity. Leveraging 103 clinical isolates, *adeG*-deleted mutants, and multi-dimensional assays, this study addresses these gaps, demonstrating that *A. baumannii* forms planktonic aggregates in backgrounds with *adeG*-related RND efflux system defects, and that these aggregates boost bacterial adaptation and virulence. These findings advance our understanding of *A. baumannii*’s survival strategies and identify novel therapeutic targets for combating recalcitrant infections.

A foundational observation of this study is the strain-specific prevalence of planktonic aggregates in *A. baumannii* clinical isolates, with a striking bias toward non-MDR strains: 13.79% of non-MDR strains formed macroscopic aggregates visible to the naked eye, compared to only 1.35% of MDR strains. This distribution closely mirrors the pattern of RND efflux system gene deletions: 93.1% of non-MDR strains harbored deletions in RND operons (*adeSR*-*adeABC*, *adeL*-*adeFGH*, or *adeN*-*adeIJK*), whereas just 17.57% of MDR strains did so. This correlation initially suggested a link between RND efflux dysfunction and aggregate formation—consistent with the well-established role of intact RND systems in mediating MDR phenotypes in *A. baumannii* [8,9]. However, our data refine this relationship by revealing that residual RND efflux activity, not mere gene deletions, is the key modulator. For example, clinically isolated strains, HFab23 (non-aggregating) and HFab43 (aggregating), shared identical Δ*adeC* genotypes, but HFab23 retained higher residual efflux activity (evidenced by lower ethidium bromide [EtBr] retention). This finding extends beyond the canonical role of RND pumps in antibiotic extrusion [32], demonstrating that these systems regulate *A. baumannii* physiology, including membrane homeostasis and intercellular adhesion, in ways that directly impact aggregate assembly. The clinical relevance of this phenotype is underscored by its presence across diverse specimens: aggregate-forming strains were isolated from cerebrospinal fluid (HFab24, a sterile site), sputum (HFab23/HFab39/HFab43/HFab104), and urine (HFab55). This confirms that planktonic aggregates are not a laboratory artifact but a real-world trait, enabling *A. baumannii* to persist in liquid-phase niches (e.g., bloodstream, respiratory tract) where surface-attached biofilms are less adaptive [3].

The AdeFGH RND efflux pump, whose functional properties remain incompletely characterized, differs from the canonical AdeABC system, in that its core gene, *adeG,* encodes a LysR-type transcription factor-regulated transmembrane channel protein, with its overexpression conferring enhanced resistance to antibiotics like tigecycline. Beyond drug efflux, AdeFGH also participates in bacterial aggregation: prior studies have shown it can modulate quorum-sensing molecule synthesis/transport to indirectly regulate biofilm formation and co-expression of *adeG,* and the quorum-sensing gene *abaI* can promote exopolymer production and bacterial clustering [33,34]. Although planktonic aggregates were formed in strain YZUMab17 following *adeG* knockout, this aggregation phenotype was not eliminated in the *adeG*-complemented strain YZUMab17Δ*adeG*. Furthermore, no planktonic aggregation was observed in YZUMab17 when the entire *adeFGH* operon was deleted. Meanwhile, ZJab1, a strain with a distinct genetic background, also failed to form planktonic aggregates after *adeG* ablation. Based on these findings, we speculate that the *adeG* gene does not directly regulate the formation of planktonic aggregates in *A. baumannii*. The planktonic aggregation phenotype induced by *adeG* knockout is strain-specific, and it is most likely attributable to the individual effects of polarity, secondary mutations, or background strain defects caused by the gene-editing procedure.

It was observed that the drug efflux activity of strain YZUMab17 was significantly reduced following *adeG* knockout, which is consistent with the phenomenon observed in clinical strains—specifically, clinical strains exhibiting the planktonic aggregation phenotype generally harbor deletions in RND efflux system genes or show decreased efflux activity. Nevertheless, the effect of *adeG* on RND efflux system function is strain-specific. In contrast to YZUMab17, ZJab1 displayed a tendency toward increased drug efflux activity after *adeG* ablation (albeit with no statistically significant difference), accompanied by upregulated expression of the *adeB* gene; additionally, the transcriptional pattern of *adeJ* in ZJab1 was completely opposite to that in YZUMab17. Given the marked genetic background differences between YZUMab17 and ZJab1, the opposite transcriptional trends of *adeJ* (upregulated in YZUMab17Δ*adeG*; downregulated in ZJab1Δ*adeG*) reflect genotype-specific regulatory differences rather than a direct modulation by *adeG* deletion. Moreover, *adeG* deletion remodeled growth kinetics to support aggregate assembly—YZUMab17Δ*adeG* displayed a prolonged lag phase and reduced maximum growth rate (likely due to the metabolic cost of adhesive structure synthesis [35]), with aggregate initiation at 6 h post-inoculation (coinciding with logarithmic growth), a pattern distinct from nutrient-limited pellicle formation [5,6]. Collectively, these findings position *adeG* as an aggregation initiator that acts via cumulative RND efflux impairment, rather than direct gene regulation, reinforcing that aggregate-mediated traits are the key clinical and mechanistic focus of this study.

At the biophysical level, successful formation of planktonian aggregates depends on the coordinated regulation of bacterial motility, surface hydrophobicity, and self-aggregation ability, and changes in these traits are associated with the altered function of the RND efflux system: The relative movement velocity (RMV) of the YZUMab17Δ*adeG* strain continued to increase after 6 h and reached the peak at 12 h, which was synchronized with the aggregate expansion process. Its “interior rough + periphery smooth” colony morphology cannot be attributed to hydrophobicity alone, and interior roughness may also be influenced by factors such as EPS secretion and local growth rate. However, combined with the high hydrophobicity data verified by MATH experiments (Figure 9C) [22,24], it can be comprehensively judged that the strain has significantly increased hydrophobicity. In addition, the self-aggregation ability of this strain was enhanced, which further ensured the stability of bacterial adhesion. Pearson correlation analysis showed that the aggregation formation was significantly positively correlated with the above three traits, which revealed that the core of aggregate assembly was the synergistic effect of multiple biophysical traits, rather than the direct regulation of a single gene. The genetic background associated with *adeG* only provides the conditions for the coordinated expression of these phenotypes, with the ultimate functional outcome being the efficient formation of planktonic aggregates.

A central finding is that planktonic aggregates enhance antibiotic tolerance independently of *adeG* deletion. While *adeG* deletion induced strain-specific MIC shifts, YZUMab17Δ*adeG* exhibited enhanced survival under antibiotic stress despite these reduced MICs—a discrepancy explained by aggregation. Survival assays revealed a gradient: aggregated YZUMab17Δ*adeG* > disaggregated YZUMab17Δ*adeG* > parental YZUMab17. Non-aggregated YZUMab17Δ*adeG* (dynamic culture) showed no survival advantage, ruling out *adeG* deletion as a tolerance driver. ATP quantification uncovered the mechanism: aggregated YZUMab17Δ*adeG* had significantly lower ATP levels than disaggregated cells and parental YZUMab17, indicating “metabolic dormancy” [28]. Slow-growing, low-ATP cells are less susceptible to antibiotics targeting active processes (e.g., meropenem for cell wall synthesis, levofloxacin for DNA replication), a conserved tolerance mechanism extended here to *A. baumannii* [36].

Planktonic aggregates also boost *A. baumannii*’s ability to evade host immunity and cause persistent infections. Serum resistance assays showed that YZUMab17Δ*adeG* maintained stable proliferation across serum concentrations (1/16 U-1/512 U), while parental YZUMab17 exhibited concentration-dependent growth decline (bacteriostatic rate > 50% at 1/64 U serum). This resistance is mediated by dense, serum-induced flocculent aggregates that shield bacteria from complement proteins and antimicrobial peptides [37], analogous to biofilm-mediated immune evasion [7] but adapted for liquid environments. In a cyclophosphamide-induced immunosuppressed mouse model (mimicking ICU patients), YZUMab17Δ*adeG* exhibited significantly persistent bacteremia. In vitro, YZUMab17Δ*adeG* showed an ~80-fold higher adhesion rate and ~100-fold higher invasion rate to A549 lung epithelial cells, attributed to aggregate-associated pili [38]. These data position aggregates as critical virulence determinants.

Temporal transcriptional analysis revealed robust *csu* operon upregulation (critical for chaperone–usher pilus assembly) as a key driver of aggregation: YZUMab17Δ*adeG* exhibited explosive *csu* activation, aligning with aggregate initiation at 6 h. In contrast, non-aggregating ZJab1Δ*adeG* lacked *csu* upregulation, and isolated *pil* activation was insufficient. Fourier transform infrared (FTIR) spectroscopy and enzyme assays confirmed proteins as the core matrix component, while polysaccharides/fatty acids were excluded due to low abundance [39]. Proteinase K disrupted aggregates in a concentration-dependent manner, whereas sodium periodate (polysaccharide degrader) and DNase I (nucleic acid degrader) had no effect. This confirms that proteins—likely *csu*-mediated pili—are essential for aggregate stability.

To clarify the regulatory network associated with *adeG* deletion, we demonstrate that *adeG* deficiency is linked to upregulation of the *csu/pil* operon, enhancing both cell surface hydrophobicity and bacterial motility, which synergistically promote the formation of protein-dependent planktonic aggregates (Figure 13). These aggregates act as a dual-functional hub: they induce ATP reduction and metabolic dormancy to boost antibiotic tolerance (e.g., against meropenem) while enriching *csu/pil*-encoded fimbriae to strengthen host–cell adhesion/invasion and immune evasion—ultimately elevating bacterial virulence. Targeting aggregate matrices (e.g., Proteinase K, pilus-specific antibodies) could sensitize *A. baumannii* to antibiotics. Inhibiting *csu* operon expression may also prevent aggregation [27]. However, future studies must validate these strategies in vivo, as Proteinase K’s host toxicity remains untested. This study has limitations: (1) other RND genes (e.g., *adeB*, *adeJ*) may contribute to aggregation, requiring further investigation; (2) signaling pathways linking *adeG* to *csu*/*pil* expression (e.g., BfmRS two-component system [37]) remain unclear; (3) animal experiments used immunosuppressed mice aggregate function in immunocompetent hosts requires exploration.

## 5. Conclusions

In this study, a mechanistic functional model of planktonic aggregate biogenesis and pathogenicity in *A. baumannii* was established. Cumulative impairments in the RND efflux system lift the repression of bacterial motility, surface hydrophobicity, and auto-aggregation, laying the foundation for intercellular adhesion and cluster formation. Subsequent robust upregulation of the *csu* operon and biosynthesis of proteinaceous matrices further reinforce aggregate integrity and stability. Functionally, these planktonic aggregates act as core adaptive and virulence units: they mediate antibiotic tolerance via metabolic quiescence, evade serum-mediated killing through physical barrier effects, and enhance pathogenicity by facilitating host–cell colonization and invasion. Taken together, these insights identify aggregate matrices as key novel therapeutic targets, furnishing a promising approach to counteract refractory *A. baumannii* infections.

## Figures and Tables

**Figure 1 microorganisms-14-00008-f001:**
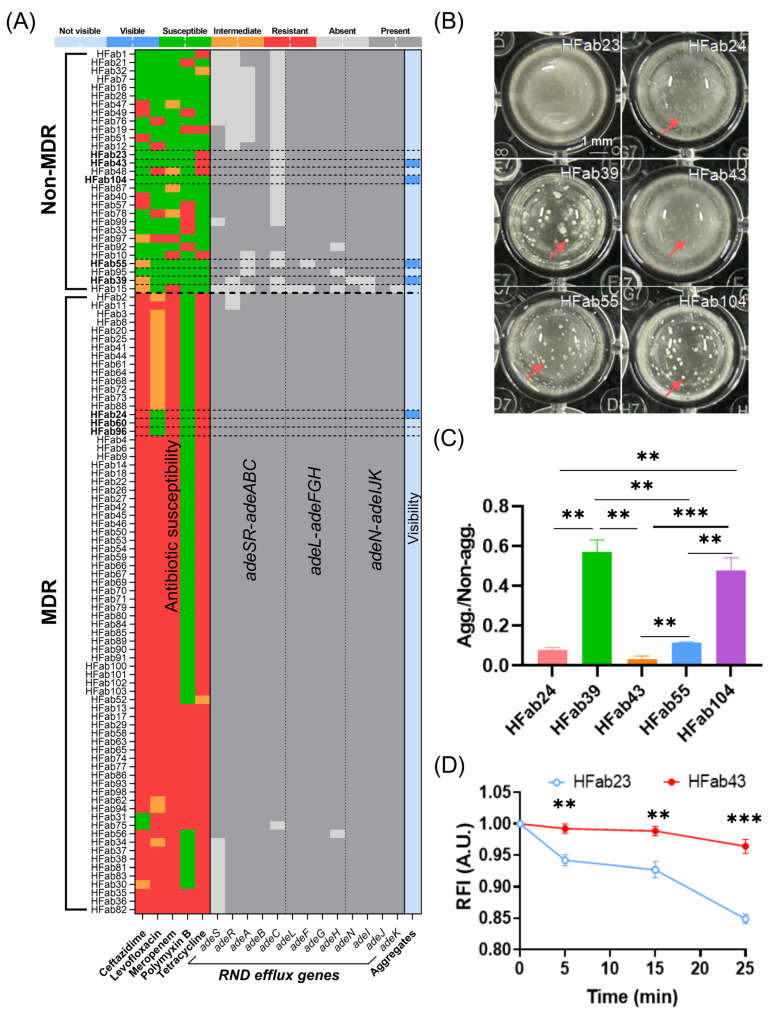
Antimicrobial resistance profile, genetic distribution, morphological characteristics, and efflux activity of planktonic aggregate-forming strains in *Acinetobacter baumannii* clinical isolates. (**A**) Heatmap of antimicrobial resistance and RND efflux pump gene distribution (breakpoints were defined for ceftazidime as ≤8, 16, and ≥32 μg/mL; for levofloxacin, as ≤2, 4, and ≥8 μg/mL; for meropenem, as ≤4, 8, and ≥16 μg/mL; for polymyxin, B as ≤ 2 and ≥ 4 μg/mL; and for tetracycline, as 4, 8, and 16 μg/mL for designating strains as antibiotic sensitive, intermediate (if available), and resistant, respectively. Multi-drug-resistant (MDR) bacteria are defined as bacteria that are resistant to at least three antibiotics.) (**B**) Visualization of planktonic aggregates formed in culture medium (the red arrow indicates the aggregate.) (**C**) Quantitative ratio of aggregated vs. non-aggregated bacterial cells in planktonic cultures. (**D**) RND efflux activity of *A. baumannii* clinical strain HFab23 versus HFab43 (EthBr efflux assay). ** and *** indicate significant difference levels of 0.01 and 0.001, respectively.

**Figure 2 microorganisms-14-00008-f002:**
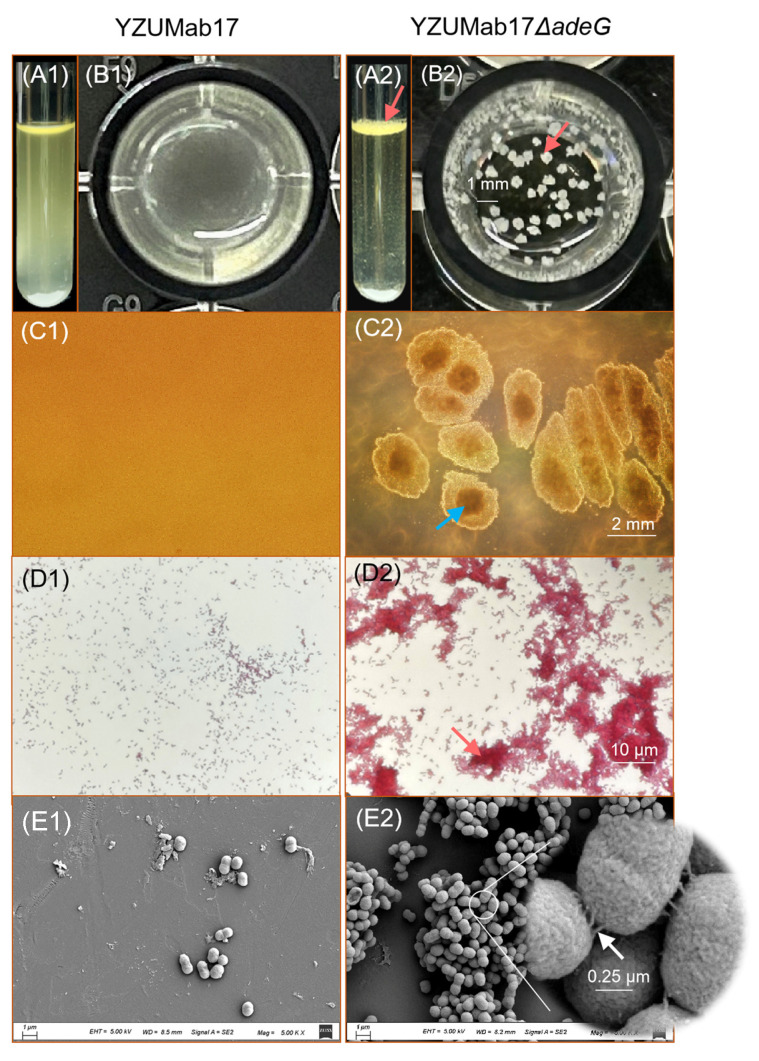
Morphological observations of *A. baumannii* YZUMab17 before and after *adeG* gene deletion. (**A1**,**A2**), lateral views of cultures in glass test tubes; (**B1**,**B2**), top–down views of cultures in 96-well microplates; (**C1**,**C2**), top–down views of cultures in Petri dishes (diameter: 10 cm); (**D1**,**D2**), Gram staining images; (**E1**,**E2**), Scanning Electron Microscopy (SEM) images. The red arrow indicates the aggregate, the blue arrow indicates the core region of the aggregate, and the white arrow indicates the filamentous adhesive substance.

**Figure 3 microorganisms-14-00008-f003:**
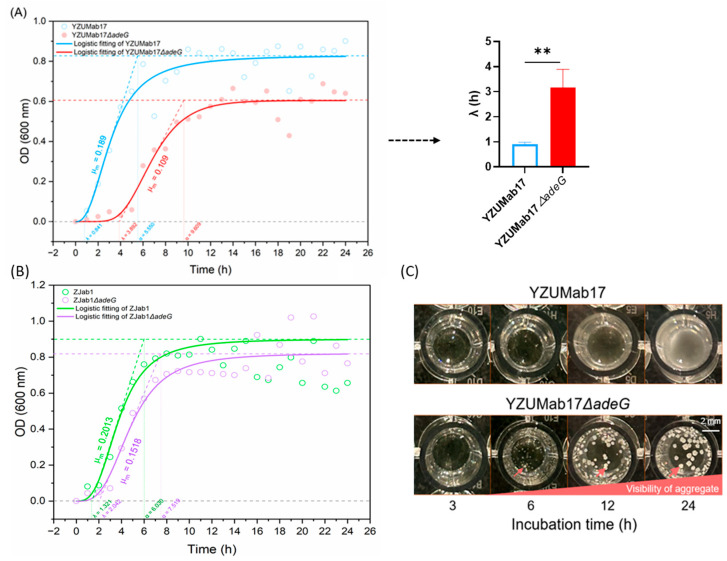
Logistic fitting of growth curves, aggregation onset observation of *A. baumannii* YZUMab17, and its *adeG*-deficient strain. (**A**,**B**) Growth curve and lag phase difference analysis under dynamic culture conditions; (**C**) phenotypic observation of aggregates under static cultivation. The dashed line indicates that this statistical plot is derived from Figure (**A**), where λ denotes the end time point of the lag phase, α denotes the end time point of the logarithmic growth phase, and μ_m_ represents the maximum growth rate. ** denotes statistical significance at the 0.01 level.

**Figure 4 microorganisms-14-00008-f004:**
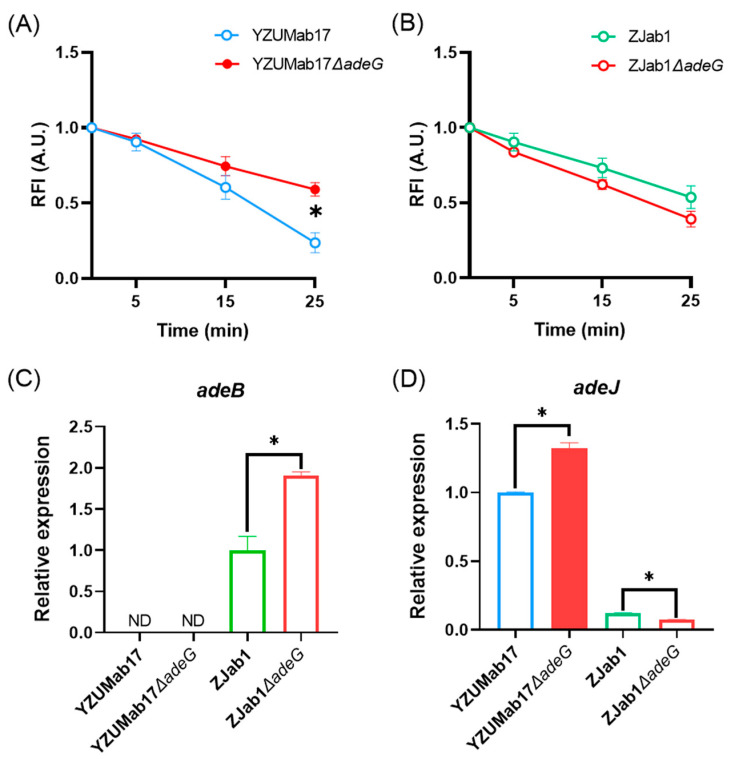
Comparison of ethidium bromide (EtBr) efflux capacity and RND efflux pump-related gene expression levels before and after *adeG* gene deletion in *Acinetobacter baumannii*. (**A**,**B**) Relative fluorescence intensity (RFI) of *A. baumannii* strains before and after *adeG* deletion as a function of time. A lower RFI value represents a higher EtBr efflux capacity, that is, a higher RND efflux pump activity. (**C**) The relative expression level of the *adeB* gene. (**D**) The relative expression level of the *adeJ* gene. ND indicates no significance; * indicates a significance level of 0.05.

**Figure 5 microorganisms-14-00008-f005:**
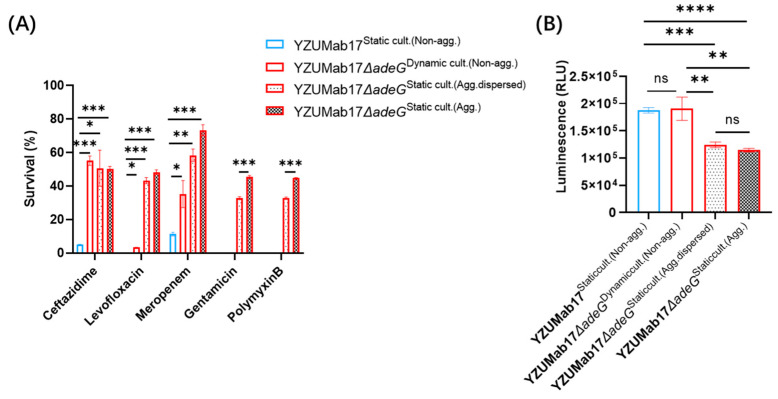
Survival dynamics and energy metabolism characteristics of *A. baumannii* YZUMab17 and its *adeG*-deficient strain under different culture conditions, aggregation phenotypes, and antibiotic stress. (**A**) Survival characteristics with different aggregation states under five antibiotics; note that ceftazidime (CAZ), levofloxacin (LEV), and meropenem (MEM) were incubated at 8× MIC for 12 h, while polymyxin B (PB) and gentamicin (GEN) were incubated at 4× MIC for 4 h. (**B**) Differences in ATP levels under dynamic/static culture and aggregated/non-aggregated states. ns indicates no significance; *, **, ***, and **** indicate significant difference levels of 0.05, 0.01, 0.001, and 0.0001, respectively.

**Figure 6 microorganisms-14-00008-f006:**
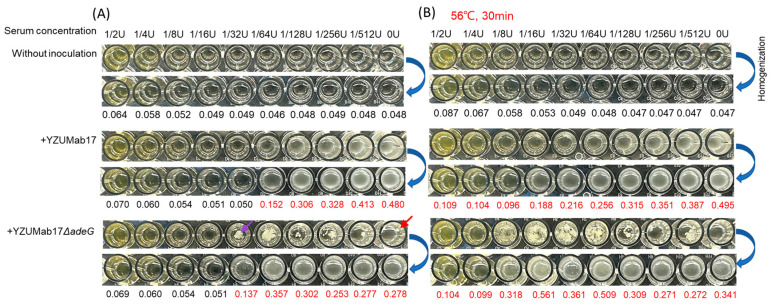
Comparison of serum bacteriostatic effects between *A. baumannii* YZUMab17 and its *adeG*-deficient strain. (**A**) Untreated serum (normal complement activity). (**B**) Inactivated serum at 56 °C for 30 min (complement inactivation). Note that each subpanel contains two rows of data with distinct purposes: the first row focuses on the qualitative observation of aggregate morphology, while the second row is for quantitative detection. Purple arrows indicate flocculent aggregates; red arrow indicates granular aggregates; values marked in red indicate significant proliferation compared with the non-inoculated group.

**Figure 7 microorganisms-14-00008-f007:**
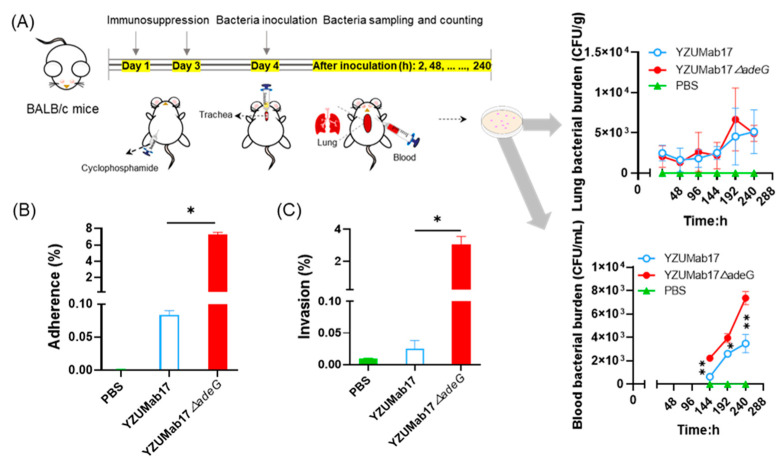
Animal infection kinetics and cellular adhesion/invasion abilities between *A. baumannii* YZUMab17 and its *adeG*-deficient strain. (**A**) In vivo infection model in mice. Bacterial burden in the lung was quantified as CFU/g lung tissue (pulmonary colonization), and systemic dissemination was assessed as CFU/mL blood (bacteremia marker), with n = 4 mice per group at each time point. (**B**) Adhesion rate to A549 cells. (**C**) Invasion rate of A549 cells. * and ** indicate significant difference levels of 0.05 and 0.01, respectively.

**Figure 8 microorganisms-14-00008-f008:**
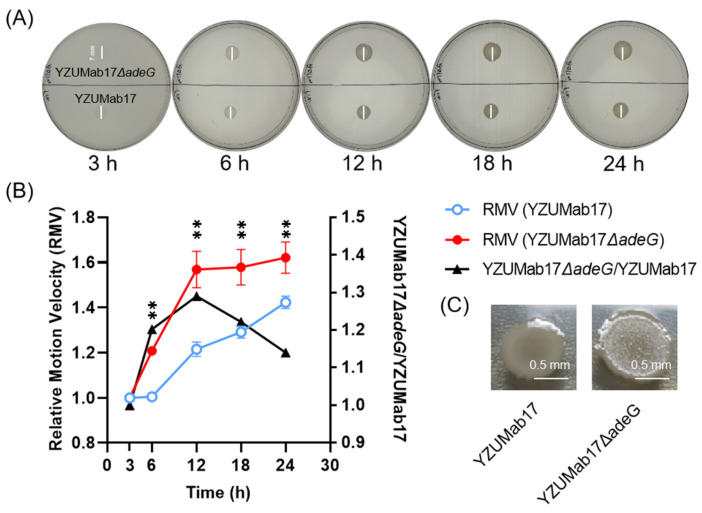
Surface motility of *Acinetobacter baumannii* strains. (**A**) Representative images of surface motility for the indicated strains assayed on semisolid LB agar plates. (**B**) Analysis of relative motility velocity (RMV) and its ratio. (**C**) Colony morphology of strains in semisolid medium. ** indicates that the relative motility velocities of the two strains are significantly different at the *p* < 0.01 level.

**Figure 9 microorganisms-14-00008-f009:**
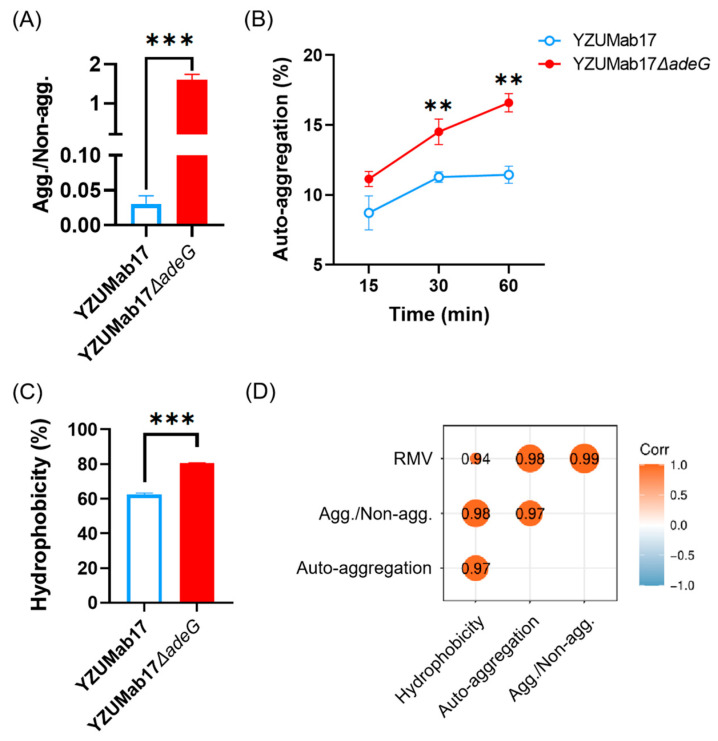
Aggregation and hydrophobicity characteristics of *A. baumannii* YZUMab17 and its *adeG*-deficient strain. (**A**) Ratio of aggregated to non-aggregated bacterial cells in planktonic cultures; (**B**) auto-aggregation rate; (**C**) hydrophobicity; (**D**) matrix graph of correlations (Corr.) among planktonic aggregate formation, auto-aggregation, hydrophobicity, and surface motility. RMV, relative motility velocity; ** and *** indicate significant differences at the levels of 0.01 and 0.001, respectively.

**Figure 10 microorganisms-14-00008-f010:**
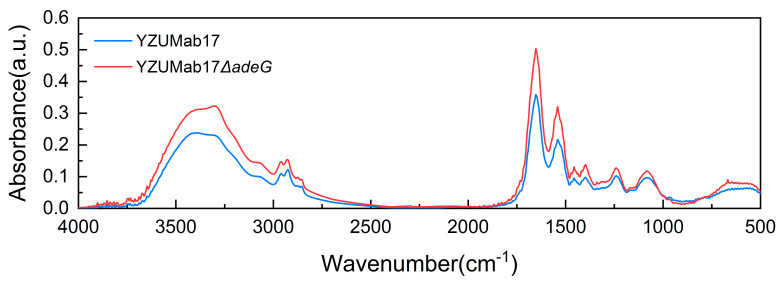
Fourier transform infrared (FTIR) spectra of planktonic aggregates formed by *A. baumannii* YZUMab17 and YZUMab17Δ*adeG*.

**Figure 11 microorganisms-14-00008-f011:**
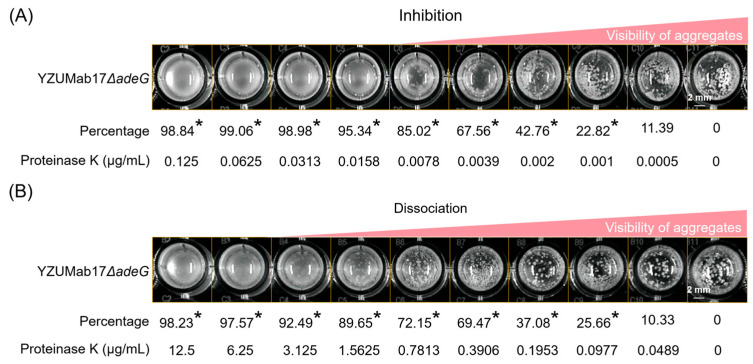
The impact of proteinase K on bacterial aggregates. (**A**) Efficiency of proteinase K in inhibiting aggregation. (**B**) Efficiency of proteinase K in dissociating pre-formed aggregates. * indicates significant differences at the level of 0.05.

**Figure 12 microorganisms-14-00008-f012:**
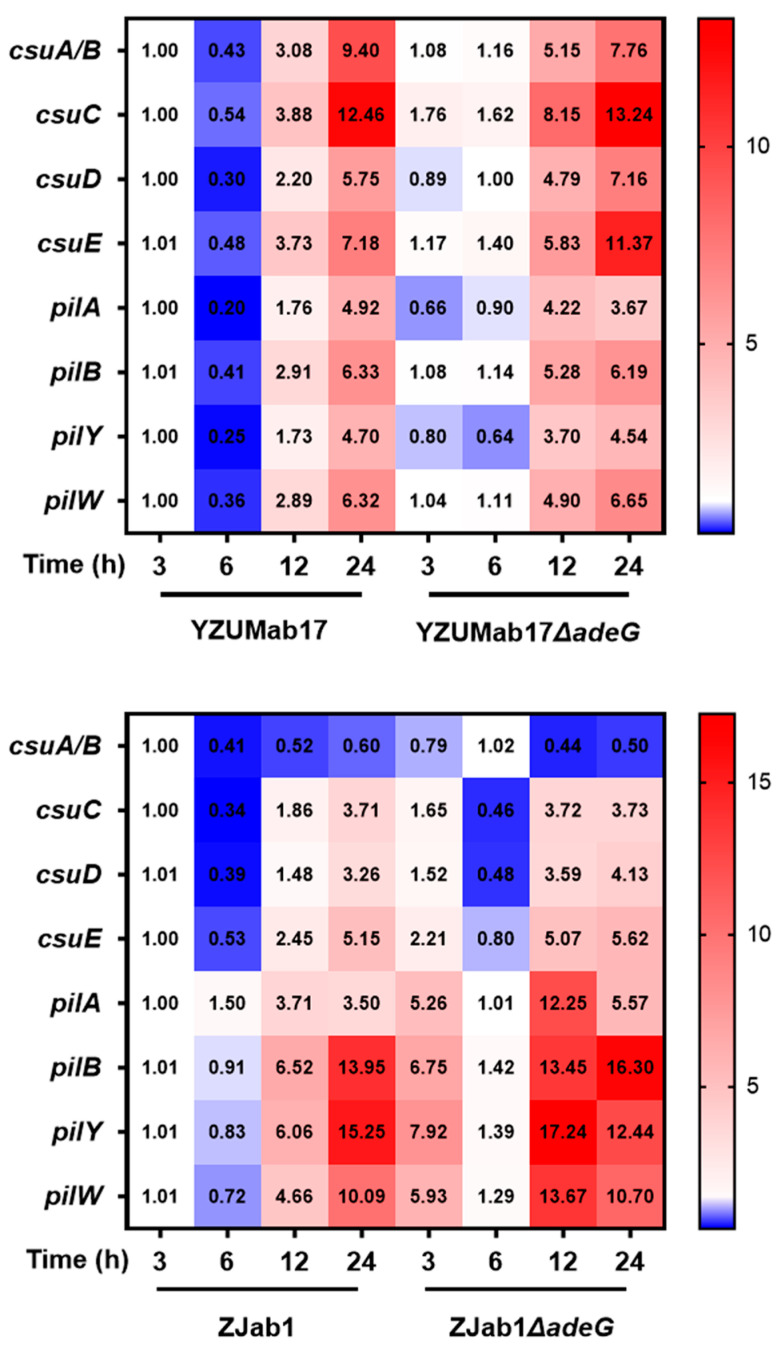
Heatmap of gene expression in *A. baumannii*. Relative expression of *csu* family and *pil* family genes at 3, 6, 12, and 24 h before and after *adeG* deletion. Darker colors (red) indicate higher expression levels, and lighter colors (blue) indicate lower expression levels (color bars indicate the corresponding expression range).

**Figure 13 microorganisms-14-00008-f013:**
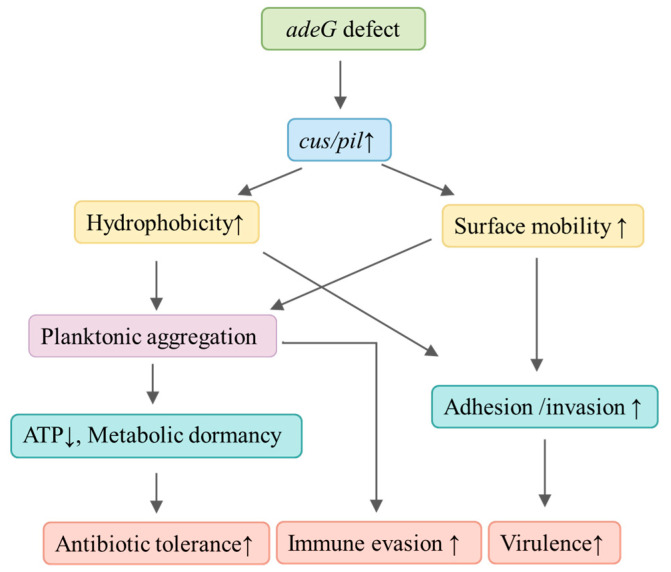
Schematic of the *adeG* deletion-mediated regulatory cascade driving aggregation, antibiotic tolerance, and virulence in non-MDR *A. baumannii*. The *adeG* deficiency enhances the hydrophobicity and surface motility of cells by upregulating the csu/pil operon and synergistically induces the formation of planktonic aggregates; the aggregates can enhance antibiotic tolerance by inducing ATP downregulation and metabolic dormancy, and simultaneously enhance adhesion/invasion ability to promote immune evasion and upregulation of virulence. ↑ indicates enhancement/upregulation; ↓ indicates reduction/downregulation.

**Table 1 microorganisms-14-00008-t001:** Minimum inhibitory concentrations (MICs) of antibiotics against *A. baumannii* (µg/mL).

Antibiotics	YZUMab17	YZUMab17Δ*adeG*	ZJab1	ZJab1Δ*adeG*
CAZ	4	2	8	8
LEV	0.25	0.125	0.25	0.125
MEM	0.25	0.125	0.125	0.125
PB	2	2	2	2

CAZ, ceftazidime; LEV, levofloxacin; MEM, meropenem; PB, polymyxin B.

**Table 2 microorganisms-14-00008-t002:** Main functional group assignments of infrared bands identified in planktonic aggregates formed by *A. baumannii* YZUMab17 and YZUMab17Δ*adeG*.

Frequency (cm^−1^)	Assignment	Relative Amount	
		YZUMab17	YYZUMab17Δ*adeG*
3410	Symmetric and asymmetric stretching of the –OH bond in water	0.664684	0.617513
2970	C–H asymmetric and symmetric stretching of methyl in fatty acids	0.281264	0.27794
2940	C–H asymmetric and symmetric stretching of methylene in fatty acids	0.295307	0.284145
1650	C=O stretching in amide I from proteins	1	1
1550	N–H bending in amide II from proteins	0.548911	0.58073
1450	C–H deformation of methylene in fatty acids	0.248134	0.237679
1400	Symmetric stretching vibration of COO^−^	0.273137	0.272592
1250	Asymmetric stretching vibration of PO_2_^−^	0.269455	0.239866
1080	C–O and C–O–C ring vibrations in polysaccharides	0.268297	0.234314

## Data Availability

The original contributions presented in this study are included in the article/Appendix A. Further inquiries can be directed to the corresponding author.

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
