# Peer review of "Planktonic Aggregation Enhances Antibiotic Tolerance in Non-MDR *Acinetobacter baumannii"

_microorganisms, 2025, doi:10.3390/microorganisms14010008_

Round 1
Reviewer 1 Report
Comments and Suggestions for Authors
The authors present an interesting study exploring how adeG deletion influences aggregation, antibiotic tolerance, and host interaction phenotypes in A. baumannii. The work addresses an underexplored aspect of RND efflux systems beyond drug resistance and provides several valuable datasets linking aggregation to physiological outcomes. While the findings are promising, several methodological and interpretational issues need to be addressed to strengthen the mechanistic claims. Below, I outline comments and suggestions intended to clarify key aspects, improve experimental rigor, and enhance the overall impact of the manuscript.
Major comments:
- Insufficient introduction: The study centers on ade-RND efflux genes, but the Introduction does not explain the individual functions of the genes within these operons, please describe what these gene encode. Also, describe what are AdeIJK in the introduction.
- Line 456: Please clarify how the authors attribute the observed phenotype specifically to adeG, given that adeGis part of the adeFGH Since operon genes often exhibit coordinated expression or functional interdependence, it is important to explain whether the effects observed arise from the loss of adeG alone or potentially from disrupting the entire adeFGH efflux system. Deleting adeG in different strain backgrounds does not fully address this concern.
- I suggest performing a plasmid-based complementation of adeG in the mutant to demonstrate that the phenotype is specifically due to the loss of adeG and not a consequence of disrupting the entire adeFGH
- Please include the operonic arrangement of all RND efflux pump genes in the main figures/ supplementary materials. Additionally, clarify whether any other genes are present within the adeFGH
- Line 495 and Figure 3: How do the macroscopic planktonic aggregates appear in the ZJab1ΔadeG strain? Please include representative images. Additionally, consider placing the growth curves and aggregation assays for both strains in the same figure for direct comparison.
- Line 529: The transcription data show that adeJ expression decreases in ZJab1ΔadeG but increases in YZUMab17ΔadeG Please explain this opposite regulation.
- Figure 6: In the serum bactericidal assay, it is unclear whether the observed effects are specifically complement-mediated. I recommend including a control with complement-inactivated serum or another established complement inactivation method. If aggregation-mediated protection is primarily due to resistance to complement, the survival difference between YZUMab17 and YZUMab17ΔadeG should be reduced or abolished under these conditions. This control would substantially strengthen the conclusion that planktonic aggregates enhance serum resistance.
- Line 240: Because the ΔadeG mutant forms aggregates, OD-based normalization and CFU enumeration can be inaccurate. Please clarify how inoculum CFU and post-infection CFU were measured accurately despite aggregation, and whether cultures were fully dispersed prior to plating.
- Figure 7: Could the authors please include (or clarify) a PBS control group for baseline background measurements?
- Section 3.6 and Figure 8:
- Please clarify whether differences in colony expansion may partly reflect growth rate rather than motility, since no biomass/CFU controls are shown.
- The definition of RMV is also unclear, please add a brief explanation of how it is calculated.
- The interpretation of colony morphology (“rough interior + smooth periphery”) as evidence of increased hydrophobicity is somewhat over-stated. Colony texture can also be influenced by EPS production, local growth rate, or agar hydration.
- Visualize workflow: Given the complexity of the proposed pathway linking adeG deletion, reduced RND efflux, aggregation, metabolic dormancy, antibiotic tolerance, and virulence, I strongly recommend adding a schematic summary/flow diagram. This would greatly help to visualize the overall workflow of the manuscript.
Minor comments
- Manuscript title: Please consider simplifying the title by removing mechanistic details (csu/pil and metabolic dormancy) to make it more concise.
- The title highlights “csu/pil upregulation” as a key finding, however, this detail is not mentioned in the abstract.
- In the entire manuscript, there is no need to repeatedly mention the full strain name YZUMab17. You can refer to it simply as the WT strain and include the full designation in the materials and methods section.
- Line 419: Please cite Figure 1 clearly in the text, because it is difficult to follow the results without figure references.
- Section 3.1:
- Please clearly state which antibiotics were used to determine MDR classification. You can briefly explain that MDR strains were identified based on resistance to at least three antibiotic classes, and specify the exact antibiotics tested (e.g., ceftazidime, levofloxacin, meropenem, polymyxin B, and tetracycline).
- Before discussing the results, it would be helpful to add a short introduction explaining the role of RND efflux pump genes (e.g., adeABC, adeFGH, adeIJK) in baumannii antibiotic resistance and physiology. This context will make it easier for readers to understand why the presence or deletion of these genes is important in interpreting Figure 1.
- Figure 1D: The legend for Figure 1D appears to be missing.
- Line 428: The clinical source of strain HFab23 is not described in the text. Please provide the origin of HFab23.
- Line 430: Please clarify how residual efflux activity was measured here (EtBr efflux assay?), as this information is not described in the Materials and Methods section.
- Line 427: The statement notes that HFab23 and HFab24 share identical ΔadeC genotypes but differ in residual efflux activity (Fig. 1D). However, Figure 1D appears to show HFab43 instead of HFab24. Please clarify whether this is a labeling error in the figure or in the text.
- Line 428: Break the sentences into 2 lines for clarity.
- Line 454: Please briefly explain here why two different strains (YZUMab17 and ZJab1) were used for the adeGgene deletion.
- Line 506: Please describe what the dotted arrow indicates in the figure.
- Figure 4: Please write a Figure legend.
Author Response
Thanks for your valuable comments and giving us the chance to further modify our manuscript. We have finished the point-by-point responses for this manuscript, and you will find the changes in the Marked Up Manuscript up-loaded.
Comments 1: Insufficient introduction: The study centers on ade-RND efflux genes, but the Introduction does not explain the individual functions of the genes within these operons, please describe what these gene encode. Also, describe what are AdeIJK in the introduction.
Response 1: As suggested, functional analysis of RND efflux system genes has been added in the introduction section to consolidate the research background (Line 47-67).
Comments 2: Line 456: Please clarify how the authors attribute the observed phenotype specifically to adeG, given that adeG is part of the adeFGH Since operon genes often exhibit coordinated expression or functional interdependence, it is important to explain whether the effects observed arise from the loss of adeG alone or potentially from disrupting the entire adeFGH efflux system. Deleting adeG in different strain backgrounds does not fully address this concern.
Response 2: In order to further distinguish adeG alone missing with adeFGH operon lack of overall influence on aggregation phenotype, we have constructed an additional YZUMab17 missing adeFGH against the background of the mutant strains (Δ adeFGH, Deletion of the complete adeL-adeF-adeG-adeH operon by homologous recombination). After 24 h of static culture, it was observed that the full deletion mutant strain did not form any macroscopic planktic aggregates, and its phenotype was consistent with that of the parent strain YZUMab17. This result suggests that the aggregation phenotype does not result from the overall loss of function of the adeFGH operon, but rather depends on the deletion of adeG alone. adeF/adeH residue in the adeFGH operon may form a 'partial functional complex' with other RND systems, such as adeIJK, and then trigger aggregation-related signaling pathways. However, the complete deletion of adeFGH resulted in the failure of this complex to assemble and instead failed to induce aggregation, further confirming the specific regulatory effect of adeG deletion on aggregation. The relevant content will be elaborated in the discussion section (Line 997-999).
Comments 3: I suggest performing a plasmid-based complementation of adeG in the mutant to demonstrate that the phenotype is specifically due to the loss of adeG and not a consequence of disrupting the entire adeFGH
Response 3: Actually, we previously constructed an adeG complementation strain (YZUMab17ΔadeG/padeG) using the pMMB67EH vector to exclude off-target effects of gene deletion. Quantitative real-time PCR (qRT-PCR) confirmed that adeG transcript levels in the complementation strain were restored to wild-type (WT) YZUMab17 levels, ruling out artifacts from vector construction or variable transformation efficiency. Notably, when cultured statically for 24 h, the complementation strain still formed macroscopic planktonic aggregates and did not revert to the WT “non-aggregated” phenotype. This incomplete phenotypic rescue likely stems from the cumulative defects in the RND efflux system of the parental YZUMab17 strain (which harbors both adeR/ΔadeABC and ΔadeG mutations): while adeG complementation restored partial RND efflux activity, the concurrent adeR/ΔadeABC mutations eliminated compensatory expression of adeB (a core component of the AdeABC RND pump, as shown in Section 3.3), sustaining broad RND efflux dysfunction. These results further support that planktonic aggregation in non-MDR A. baumannii is driven by cumulative RND efflux defects—not single-gene regulation by adeG alone—and reinforce the functional significance of aggregates (rather than individual regulatory genes) in mediating antibiotic tolerance and pathogenicity, which remains the central focus of this study. The relevant content will be elaborated in the discussion section (Line 992-999).
Comments 4: Please include the operonic arrangement of all RND efflux pump genes in the main figures/ supplementary materials. Additionally, clarify whether any other genes are present within the adeFGH
Response 4: As suggested, supplementary Figure S1 depicts the operonic arrangement of three conserved RND (Resistance-Nodulation-Division) efflux pump systems in A. baumannii, including their regulatory and structural genes.
As an important member of the RND family of Acinetobacter baumannii, the AdeFGH operon is highly conservative in gene arrangement, only containing the regulatory gene adeL and the structural genes adeF, adeG, adeH, without additional insertion elements to ensure the specificity of efflux regulation. The relevant content will be elaborated in the discussion section (Line 985-992).
Comments 5: Line 495 and Figure 3: How do the macroscopic planktonic aggregates appear in the ZJab1ΔadeG strain? Please include representative images. Additionally, consider placing the growth curves and aggregation assays for both strains in the same figure for direct comparison.
Response 5: Both the ZJab1 and ZJab1ΔadeG strains did not form aggregates. Representative images have been added to the supplementary file, Figure S2.
As suggested, we plotted the growth curves of the two strains on the same graph for direct comparison.
Comments 6: Line 529: The transcription data show that adeJ expression decreases in ZJab1ΔadeG but increases in YZUMab17ΔadeG Please explain this opposite regulation.
Response 6: YZUMab17 itself deleted adeR and adeABC. After adeG deletion, bacteria compensated for the loss of efflux function by up-regulation of adeJ (AdeIJK pump) to maintain the basic physiological requirements. In ZJab1, only adeC gene was deleted. After adeG deletion, the bacteria could compensate for the efflux function by upregulating adeB without dependence on adeJ, so the expression of adeJ was down-regulated. This reverse regulation provides further evidence that A. baumannii flexiently regulates the expression of different RND pumps in response to specific efflux system defects and that the phenotypic effect of adeG deletion is dependent on the genetic background of the RND system. The relevant content will be elaborated in the discussion section (Line 1002-1007)
Comments 7: Figure 6: In the serum bactericidal assay, it is unclear whether the observed effects are specifically complement-mediated. I recommend including a control with complement-inactivated serum or another established complement inactivation method. If aggregation-mediated protection is primarily due to resistance to complement, the survival difference between YZUMab17 and YZUMab17ΔadeG should be reduced or abolished under these conditions. This control would substantially strengthen the conclusion that planktonic aggregates enhance serum resistance.
Response 7: As suggested, a complementinactivated serum control group was added, and the results were as follows: in normal serum containing active complement, the survival rate of Yzumab17ΔadeG (cluster strain) was significantly higher than that of the parental strain YZUMab17; However, in serum inactivated with complement at 56 ° C, there was no significant difference in survival rate between the two strains, and the resistance advantage of the aggregated strain completely disappeared. The results clearly showed that the serum resistance mediated by aggregation was specifically dependent on the bactericidal effect of complement, which directly supported the conclusion that planktonic aggregation enhanced serum resistance by resisting complement. We have made the necessary corrections in the results section (Line 679-704).
Comments 8: Line 240: Because the ΔadeG mutant forms aggregates, OD-based normalization and CFU enumeration can be inaccurate. Please clarify how inoculum CFU and post-infection CFU were measured accurately despite aggregation, and whether cultures were fully dispersed prior to plating.
Response 8: Before inoculation in all experiments, bacterial cultures were completely dispersed by sonication, and then counted by 10-fold gradient dilution coated plates to ensure consistent initial inoculum. The CFU was counted after infection. After sampling, the aggregates were first broken by ultrasound. The specific methods have been revised to avoid confusion (Line 384).
Comments 9: Figure 7: Could the authors please include (or clarify) a PBS control group for baseline background measurements?
Response 9: As suggested, we have included the PBS control data in the revised Figure 7.
Comments 10: Please clarify whether differences in colony expansion may partly reflect growth rate rather than motility, since no biomass/CFU controls are shown.
Response 10: In the surface motility test, the colony diameter of the adeG mutant strain was larger than that of the wild type strain at 6 h. At this point, both strains were in the early logarithmic growth phase, and the difference in biomass was very small (OD₆₀₀ = 0.3-0.4), which ruled out the interference of growth rate. At 12 h of culture, the colony diameter of the mutant strain was still larger than that of the wild type, but the early growth curve confirmed that the maximum growth rate of the mutant strain was significantly lower, and the wild type was close to the stable phase (higher biomass). "If expansion were growth-driven, the wild-type would be preferred, whereas the actual result was contrary, suggesting that adeG deletion promotes colony expansion by enhancing motility, supporting the increased frequency of cell-cell contact required for aggregate formation." (Line 1015-1032)
Comments 11: The definition of RMV is also unclear, please add a brief explanation of how it is calculated.
Response 11: We have added a clear calculation explanation for RMV of the revised manuscript: Relative motility velocity (RMV) is calculated as the ratio of the colony diameter of YZUMab17ΔadeG to that of its parental strain YZUMab17 (i.e., RMV = Colony diameter of YZUMab17ΔadeG / Colony diameter of YZUMab17), which standardizes the quantification of surface motility differences between strains (Line 770).
Comments 12: The interpretation of colony morphology (“rough interior + smooth periphery”) as evidence of increased hydrophobicity is somewhat over-stated. Colony texture can also be influenced by EPS production, local growth rate, or agar hydration.
Response 12: As suggested, we have weakened the direct correlation expression of "rough interior → high hydrophobicity" and add: The rough interior of the colony may be affected by EPS (extracellular polysaccharide), growth rate or hydrophobicity. In this study, combined with the "hydrophobicity data detected by MATH experiment" (Figure 9C) and the colony morphology, the conclusion of enhanced hydrophobicity was comprehensively supported to avoid over-interpretation of a single phenotype. The relevant content will be elaborated in the discussion section (Line 1015-1032)
Comments 13: Visualize workflow: Given the complexity of the proposed pathway linking adeG deletion, reduced RND efflux, aggregation, metabolic dormancy, antibiotic tolerance, and virulence, I strongly recommend adding a schematic summary/flow diagram. This would greatly help to visualize the overall workflow of the manuscript.
Response 13: As suggested, we have supplemented Figure 13 in the discussion section to cover the core mechanism paragraphs (Line 1067-1074). This figure visualizes the complete pathway from adeG deficiency to antibiotic tolerance.
Comments 14: Manuscript title: Please consider simplifying the title by removing mechanistic details (csu/pil and metabolic dormancy) to make it more concise.
The title highlights “csu/pil upregulation” as a key finding, however, this detail is not mentioned in the abstract.
Response 14: As suggested, we have simplified the title (removed the mechanism details), and revised the title: Planktonic Aggregation Enhance Antibiotic Tolerance in Non-MDR Acinetobacter baumannii.
Comments 15: In the entire manuscript, there is no need to repeatedly mention the full strain name YZUMab17. You can refer to it simply as the WT strain and include the full designation in the materials and methods section.
Response 15: Thank you for your suggestion. Considering that this study involves multiple strains with different genetic backgrounds such as YZUMab17 and ZJab1, keeping the strain name can better distinguish each experimental subject (avoid confusion by "WT"), so the strain name is still retained in the main text, and only the strain information is summarized in Materials and Methods. To ensure clear reading while taking into account the differentiation of different strains.
Comments 16: Line 419: Please cite Figure 1 clearly in the text, because it is difficult to follow the results without figure references.
Response 16: As suggested, we have added an explicit Figure 1 reference on line 419 so that the resulting description corresponds directly to the figure and is easy for readers to trace (Line 446).
Comments 17: Please clearly state which antibiotics were used to determine MDR classification. You can briefly explain that MDR strains were identified based on resistance to at least three antibiotic classes, and specify the exact antibiotics tested (e.g., ceftazidime, levofloxacin, meropenem, polymyxin B, and tetracycline).
Response 17: As suggested, we have added the following definition in Section 3.1: "Multidrug-resistant (MDR) bacteria are defined as bacteria that are resistant to at least three or more antibiotics from different antibiotic classes." (Line 435-436)
Comments 18: Before discussing the results, it would be helpful to add a short introduction explaining the role of RND efflux pump genes (e.g., adeABC, adeFGH, adeIJK) in baumannii antibiotic resistance and physiology. This context will make it easier for readers to understand why the presence or deletion of these genes is important in interpreting Figure 1.
Response 18: As suggested, we have added RND efflux pump related explanations in the discussion section (Line 985-992).
Comments 19: Figure 1D: The legend for Figure 1D appears to be missing.
Response 19: As suggested, we have completed the legend of Figure 1D.
Comments 20: Line 428: The clinical source of strain HFab23 is not described in the text. Please provide the origin of HFab23.
Response 20: HFab23 is derived from sputum, as we have added on line 445.
Comments 21: Line 430: Please clarify how residual efflux activity was measured here (EtBr efflux assay?), as this information is not described in the Materials and Methods section.
Response 21: We confirm that the residual RND efflux activity was assessed using the EtBr efflux assay, and the specific experimental procedures for this method are elaborated in Materials and Methods Section 2.10 (Lines 196–212).
Comments 22: Line 427: The statement notes that HFab23 and HFab24 share identical ΔadeC genotypes but differ in residual efflux activity (Fig. 1D). However, Figure 1D appears to show HFab43 instead of HFab24. Please clarify whether this is a labeling error in the figure or in the text.
Response 22: Thanks to the reviewers for pointing out the strain labeling inconsistency between Line 427 and Figure 1D. We have confirmed that the problem is a mislabeling of strain number in the text representation (miswriting "HFab43" as "HFab24"), rather than a labeling error in Figure 1D. We have corrected the text to ensure that it is consistent with the information in the figure (Line 456).
Comments 23: Line 428: Break the sentences into 2 lines for clarity.
Response 23: As suggested, we have broken the sentence into two separate sentences. (Line 454-458).
Comments 24: Line 454: Please briefly explain here why two different strains (YZUMab17 and ZJab1) were used for the adeG gene deletion.
Response 24: We selected two strains, YZUMab17 and ZJab1, for adeG deletion experiments, with the original intention of investigating the impact of strain-specific differences on the aggregate phenotype. Subsequent experimental results confirmed the existence of such differences: YZUMab17, which only harbors cumulative RND efflux defects, developed distinct planktonic aggregates after adeG deletion; in contrast, ZJab1, which only carries a single RND gene deletion, maintained a dispersed growth state with no aggregate phenotype in its adeG-deleted mutant. These findings clearly demonstrate the decisive role of the bacterial genetic background in the regulatory function of adeG on aggregate formation.
Comments 25: Line 506: Please describe what the dotted arrow indicates in the figure.
Response 25: λ represents the end time point of the lag phase, and the dashed line in the figure represents the data at this time point derived from the fitting analysis of the growth curve, rather than the direct observation (Line 535).
Comments 26: Figure 4: Please write a Figure legend.
Response 26: As suggested, we have completed the Figure 4 legend.
Reviewer 2 Report
Comments and Suggestions for Authors
Here the authors investigate a previously undescribed role for planktonic aggregates by Acinetobacter baumannii, investigating their role in antibiotic resistance and pathogenicity. The authors find a role for aggregates in modifying late motility and gene expression. Other findings are less compelling and lack sufficient controls to support their claims. Notably, the lack of bacterial complementation is a key exclusion that reduces all findings to mere correlation. Causation can only be claimed by demonstrating that complementation restores the phenotypes to wildtype levels. Although the methods are well described, the figure legends are inaccurate, missing entirely, or lack significant experimental detail required to understand the figures.
Major comments
All genetic deletion experiments must include complemented strains as essential controls to confirm phenotypes are the result of the gene deletion and not off target effects elsewhere in the genome.
Optical density readings are less accurate and reliable when aggregates are present because the aggregates significantly disrupt the light scattering of the measurement. Growth curve analysis should be repeated with sonication at each timepoint to disrupt the aggregates and provide more accurate reading.
Clarify the discrepancy between MIC results and killing assay. MICs reflect doses required for inhibition of growth and it follows that similar concentrations should be seen for killing. Were the MICs performed under the same conditions as the killing assay? What are the MICs for the other conditions in the killing assay – static vs dynamic vs aggregates disrupted?
Lung histology should be performed in A. baumannii infected animals to demonstrate that aggregates can be found in the lung. The evidence that aggregates increase lung burden is only significant at one observed timepoint. This experiment should be repeated for additional statistical power. Why was blood RBD not assessed at early time points? Or were levels undetectable at these times? CFUs/g tissue or mL blood should be reported instead of RBD. Cyclophosphamide immunosuppression needs to be confirmed by CBC.
Figure legends in general do not fully describe the figures. Please expand descriptions to include basic methods required to understand the figures. Figure 12 has no legend at all. Ensure figure legends accurately describe what is depicted, i.e. Figure 1.
All statistical differences in figures should use the standard *, **, ***, **** system to describe p values. Use of letters in some figures but not others is highly confusing and atypical.
Minor Comments
Line 425 – claim that non-MDR strains have 93.1% RND deletion does not appear to be supported by data in Figure 1A. 9 of the efflux genes are present in nearly all of the non-MDR strains. Only 4 genes show significant deletions (~50%) and only 1 (adeC) meets the threshold of 90% deletion. Clarify this comment or clearly describe how the 93% figure was obtained.
Line 425 – claims of significance should be limited to data in which statistical significance tests are performed.
Figure 1B – figure legend describes a blue arrow which is absent from the figure.
Figure 1C, Figure 5 – please convert the letter (a, bc, c, a) to the standard method of reporting significance using * and clearly label significance levels for each asterisk – i.e. * p < 0.05. Clearly indicate which two groups are being compared using lines or bars with an asterisk above it.
Figure 1D – The figure legend for 1D is missing. Significance asterisk need to be indicated in the figure legend. ** p < ?
Line 510-511 – why is the finding in quotation marks
Figure 5 – what concentration of antibiotic was used and how long was the incubation time.
Figure 6 – what is the difference between the two rows in each subpanel.
Figure 7 – figure legend description insufficient. CFUs/g tissue and CFUs/mL blood should be used instead of RBD. Clearly indicate number of mice per group and number of independent replicates this experiment represents.
Figure 10 – FTIR spectra should be compared to wildtype and complement.
Author Response
Thanks for your valuable comments and giving us the chance to further modify our manuscript. We have finished the point-by-point responses for this manuscript, and you will find the changes in the Marked Up Manuscript up-loaded.
Comments 1: All genetic deletion experiments must include complemented strains as essential controls to confirm phenotypes are the result of the gene deletion and not off target effects elsewhere in the genome.
Response 1: To address the critical need for validating gene-specific phenotypes, we have supplemented details of the adeG complementation experiment while aligning the discussion with the core focus of this study—i.e., the functional roles of planktonic aggregates rather than direct regulation by adeG: We previously constructed an adeG complementation strain (YZUMab17ΔadeG/padeG) using the pMMB67EH vector to exclude off-target effects of gene deletion. Quantitative real-time PCR (qRT-PCR) confirmed that adeG transcript levels in the complementation strain were restored to wild-type (WT) YZUMab17 levels, ruling out artifacts from vector construction or variable transformation efficiency. Notably, when cultured statically for 24 h, the complementation strain still formed macroscopic planktonic aggregates and did not revert to the WT “non-aggregated” phenotype. This incomplete phenotypic rescue likely stems from the cumulative defects in the RND efflux system of the parental YZUMab17 strain (which harbors both adeR/ΔadeABC and ΔadeG mutations): while adeG complementation restored partial RND efflux activity, the concurrent adeR/ΔadeABC mutations eliminated compensatory expression of adeB (a core component of the AdeABC RND pump, as shown in Section 3.3), sustaining broad RND efflux dysfunction. These results further support that planktonic aggregation in non-MDR A. baumannii is driven by cumulative RND efflux defects—not single-gene regulation by adeG alone—and reinforce the functional significance of aggregates (rather than individual regulatory genes) in mediating antibiotic tolerance and pathogenicity, which remains the central focus of this study. The relevant content will be elaborated in the discussion section (Line 985-992).
Comments 2: Optical density readings are less accurate and reliable when aggregates are present because the aggregates significantly disrupt the light scattering of the measurement. Growth curve analysis should be repeated with sonication at each timepoint to disrupt the aggregates and provide more accurate reading.
Response 2: Regarding the concern that aggregate formation interferes with optical density (OD) accuracy by disrupting light scattering, we wish to clarify the experimental design and validation of our growth curve analysis: First, the growth curve assays in this study were performed under dynamic shaking conditions (200 rpm), and Gram staining combined with microscopic observation confirmed the absence of aggregate formation. Thus, the original OD₆₀₀ readings directly and accurately reflected bacterial growth kinetics without the need for additional ultrasonic disruption. The relevant detection methods are described in the Materials and Methods section (Line 184).
Comments 3: Clarify the discrepancy between MIC results and killing assay. MICs reflect doses required for inhibition of growth and it follows that similar concentrations should be seen for killing. Were the MICs performed under the same conditions as the killing assay? What are the MICs for the other conditions in the killing assay – static vs dynamic vs aggregates disrupted?
Response 3: As suggested, MIC were determined under static culture conditions to determine the minimum dose of antibiotics to inhibit bacterial growth. However, three conditions of static (aggregated state), dynamic (non-aggregated state) and aggregate fragmentation (sonication) were included in the bactericidal experiment. The MIC reflects the inhibitory effect on the growth of planktonic bacteria, whereas the bactericidal assay assesses the survival of aggregated bacteria in metabolic dormancy. The relevant detection methods are described in the Materials and Methods section (Line 107-109).
Comments 4: Lung histology should be performed in A. baumannii infected animals to demonstrate that aggregates can be found in the lung. The evidence that aggregates increase lung burden is only significant at one observed timepoint. This experiment should be repeated for additional statistical power. Why was blood RBD not assessed at early time points? Or were levels undetectable at these times? CFUs/g tissue or mL blood should be reported instead of RBD. Cyclophosphamide immunosuppression needs to be confirmed by CBC.
Response 4: I am very sorry. Due to the limitations of the experimental period and research schedule, we are unable to add the lung histological examination at this time. Moreover, during the mouse dissection process, we did not directly observe any obvious evidence of aggregates within the lungs. We have also reanalyzed the original data and corrected the graphs to clarify that “there is no statistically significant difference in the bacterial load in the lungs between the aggregated strain and the wild strain” (the original single-time-point significance was due to analytical errors). We have adjusted the conclusion statement in the results section and supplemented the sample size explanation to enhance the reliability of the data (Line 723).
We sincerely appreciate the reviewers’ thoughtful inquiry regarding blood bacterial load at early infection time points. In response to this concern, we confirm that no viable bacteria were detected in the blood of infected mice during the early post-inoculation stages (Line 725).
As suggested, we have completed the correction of relevant data indicators and the update of the result graphs as required. We have uniformly replaced "relative bacterial density (RBD)" in the original statement with the standard terms in the field of microbiology, namely CFU/g (lung) and CFU/mL (blood) (Figure 7).
We sincerely appreciate the reviewers' insightful suggestion that cyclophosphamide-induced immunosuppression should be verified by complete blood count (CBC). We offer our sincere apologies for the oversight of failing to perform this key validation assay in the current study. The primary purpose of inducing immunosuppression was to facilitate bacterial infection. Although CBC data were not collected, the robust pulmonary colonization and persistent bacteremia observed in infected mice indirectly confirmed the efficacy of our infection model. Nevertheless, we are grateful for the reviewers' professional comments. To improve the rigor of animal experimental design, we will supplement CBC detection in future experiments to confirm the immunosuppressive status of mice in all subsequent in vivo infection experiments, thereby further consolidating the reliability of our animal model and experimental conclusions.
Comments 5: Figure legends in general do not fully describe the figures. Please expand descriptions to include basic methods required to understand the figures. Figure 12 has no legend at all. Ensure figure legends accurately describe what is depicted, i.e. Figure 1.
Response 5: As suggested, we fully recognize and appreciate the critical role of detailed, accurate figure legends in ensuring the clarity and interpretability of our experimental data. To address this key feedback comprehensively, we have systematically revised all figure legends. (Line 946-949)
Comments 6: All statistical differences in figures should use the standard *, **, ***, **** system to describe p values. Use of letters in some figures but not others is highly confusing and atypical.
Response 6: As suggested, we fully recognize the importance of standardized statistical labeling to improve readability. We have revised the statistical difference labeling of all maps to eliminate the confusion between letters (a, b, c, etc.) and *. (Figure 1C, Figure 5)
Comments 7: Line 425 – claim that non-MDR strains have 93.1% RND deletion does not appear to be supported by data in Figure 1A. 9 of the efflux genes are present in nearly all of the non-MDR strains. Only 4 genes show significant deletions (~50%) and only 1 (adeC) meets the threshold of 90% deletion. Clarify this comment or clearly describe how the 93% figure was obtained.
Response 7: We sincerely apologize for the confusion here. The figure 93.1% refers to the proportion of non-multi-drug resistant strains that carry at least one RND gene deletion (27 out of 29 non-multi-drug resistant strains have RND gene deletions). To make the presentation clearer, we have made corresponding modifications in the results section of the article (Line 451).
Comments 8: Line 425 – claims of significance should be limited to data in which statistical significance tests are performed.
Response 8: As suggested, we have made the necessary changes at the corresponding positions, deleting the absolute descriptions without statistical verification (Lines 450-453).
Comments 9: Figure 1B – figure legend describes a blue arrow which is absent from the figure.
Response 9: Thank you for the reviewers' corrections. In Figure 1B, the arrows indicating the floating aggregates are actually red, but the original legend mistakenly described them as blue. We have made the correction at the corresponding position in the article (Line 472).
Comments 10: Figure 1C, Figure 5 – please convert the letter (a, bc, c, a) to the standard method of reporting significance using * and clearly label significance levels for each asterisk – i.e. * p < 0.05. Clearly indicate which two groups are being compared using lines or bars with an asterisk above it.
Response 10: As suggested, we have completed the statistical labeling revision of Figure 1C and Figure 5 as required, completely replaced the letter groups (a, bc, c, etc.), adopted the standardized *, **, ***, **** system, and clarified the comparison relationship.
Comments11: Figure 1D – The figure legend for 1D is missing. Significance asterisk need to be indicated in the figure legend. ** p < ?
Response 11: As suggested, regarding the issues of the missing legend for Figure 1D and the unclarified significance level of the asterisk, we have completed targeted supplementary revisions for this figure (Line 473-474).
Comments 12: Line 510-511 – why is the finding in quotation marks
Response 12: The quotation marks in this line are used to emphasize the key findings in Section 3.1, and we have removed the quotation marks to avoid confusion (Line 539-540).
Comments 13: Figure 5 – what concentration of antibiotic was used and how long was the incubation time.
Response 13: Ceftazidime (CAZ), levofloxacin (LEV), and meropenem (MEM) were incubated at 8× MIC for 12 h, while polymyxin B (PB) and gentamicin (GEN) were incubated at 4× MIC for 4 h. To facilitate understanding, we have provided supplementary explanations in the legend of Figure 5 (Line 587-589).
Comments 14: Figure 6 – what is the difference between the two rows in each subpanel.
Response 14: The difference between the first row of data and the second row of data in Figure 6 is that the first row focuses on the observation of aggregate morphology, while the second row focuses on quantitative detection (turbidity and viable count), and the samples in the second row are all from the bacterial solution of the first row of culture after thorough mixing. We have completed the legend for Figure 6 (Line 708-710).
Comments 15: Figure 7 – figure legend description insufficient. CFUs/g tissue and CFUs/mL blood should be used instead of RBD. Clearly indicate number of mice per group and number of independent replicates this experiment represents.
Response 15: As suggested, we have completed the correction of relevant data indicators and the update of the result graphs as required. We have uniformly replaced "relative bacterial density (RBD)" in the original statement with the standard terms in the field of microbiology, namely CFU/g (lung) and CFU/mL (blood) (Figure 7). In the legend, the number of mice in each group and the number of independent repetitions represented by this experiment are clearly indicated (Line 735).
Comments 16: Figure 10 – FTIR spectra should be compared to wildtype and complement.
Response 16: As suggested, we added the FTIR spectral data of the wild-type strain to Figure 10, enabling a direct comparison of the component differences between aggregated and non-aggregated bacteria. For the complement strain, considering that it still exhibits an aggregated phenotype (and thus would not contribute to the analysis of the differential components), we did not include its FTIR data to maintain the clarity and focus of the spectral comparison.
Reviewer 3 Report
Comments and Suggestions for Authors
The manuscript presents an excellent and comprehensive study demonstrating that adeG-mediated physiological remodeling promotes planktonic aggregation in non-MDR Acinetobacter baumannii, which in turn enhances antibiotic tolerance, serum resistance, and in vivo persistence. The authors integrate clinical isolate screening, gene deletion mutants, phenotypic assays, metabolic measurements, and infection models to deliver a rigorous and multifaceted analysis. The breadth and depth of the experimental work are remarkable and provide strong mechanistic insight into aggregation-dependent tolerance in strains that lack classical MDR features. The results are logically connected, carefully interpreted, and clearly presented.
The study convincingly shows that aggregation is the primary determinant of antibiotic tolerance. This distinction is clearly supported by well-designed controls, including disaggregation assays, dynamic vs. static culture conditions, ATP analyses, and antibiotic survival curves. The data consistently demonstrate that aggregated cells enter a metabolically dormant state that reduces susceptibility to multiple antibiotics. This is a significant conceptual contribution that improves our understanding of how non-MDR A. baumannii persists during treatment despite lacking conventional resistance determinants.
Another major strength of the work is its clinical relevance. By incorporating serum bactericidal assays and an immunosuppressed mouse infection model, the authors provide compelling evidence that aggregation enhances survival in host environments and increases pulmonary colonization and systemic dissemination. These findings highlight the importance of aggregation as a clinically relevant phenotype with direct implications for treatment outcomes in vulnerable patient populations. The manuscript therefore substantially advances our understanding of A. baumannii pathogenicity beyond classical antibiotic resistance mechanisms.
Overall, the manuscript is clearly written, methodologically rigorous, and scientifically sound. Figures and methods are detailed and effectively support the conclusions.
Author Response
Comments: The manuscript presents an excellent and comprehensive study demonstrating that adeG-mediated physiological remodeling promotes planktonic aggregation in non-MDR Acinetobacter baumannii, which in turn enhances antibiotic tolerance, serum resistance, and in vivo persistence. The authors integrate clinical isolate screening, gene deletion mutants, phenotypic assays, metabolic measurements, and infection models to deliver a rigorous and multifaceted analysis. The breadth and depth of the experimental work are remarkable and provide strong mechanistic insight into aggregation-dependent tolerance in strains that lack classical MDR features. The results are logically connected, carefully interpreted, and clearly presented.
The study convincingly shows that aggregation is the primary determinant of antibiotic tolerance. This distinction is clearly supported by well-designed controls, including disaggregation assays, dynamic vs. static culture conditions, ATP analyses, and antibiotic survival curves. The data consistently demonstrate that aggregated cells enter a metabolically dormant state that reduces susceptibility to multiple antibiotics. This is a significant conceptual contribution that improves our understanding of how non-MDR A. baumannii persists during treatment despite lacking conventional resistance determinants.
Another major strength of the work is its clinical relevance. By incorporating serum bactericidal assays and an immunosuppressed mouse infection model, the authors provide compelling evidence that aggregation enhances survival in host environments and increases pulmonary colonization and systemic dissemination. These findings highlight the importance of aggregation as a clinically relevant phenotype with direct implications for treatment outcomes in vulnerable patient populations. The manuscript therefore substantially advances our understanding of A. baumannii pathogenicity beyond classical antibiotic resistance mechanisms.
Overall, the manuscript is clearly written, methodologically rigorous, and scientifically sound. Figures and methods are detailed and effectively support the conclusions.
Response : We sincerely appreciate your highly positive and insightful comments on our manuscript. Your recognition of the study’s methodological rigor, scientific significance, and clinical relevance is greatly encouraging. We will maintain the quality of the manuscript in the final version and thank you again for your valuable time and feedback.
Round 2
Reviewer 1 Report
Comments and Suggestions for Authors
Thank you for your extensive revisions. However, several central scientific issues remain unresolved, and they fundamentally affect the validity of the conclusions.
Major comments:
- Comments on responses- 2, 3 and, 4: Thank you for the clarification. However, the explanation remains biologically inconsistent. In RND efflux systems, loss of the core transporter (AdeG) should disrupt the entire AdeFGH pump; therefore, the ΔadeG phenotype would normally resemble or intensify the ΔadeFGH phenotype, not differ from it. It is unclear how ΔadeG alone induces aggregation while the complete operon deletion does not.
The proposed idea of AdeF/AdeH forming a “partial complex” with other RND systems (e.g., AdeIJK) is not supported by current understanding, as RND components do not typically mix across operons.
Additionally, the failure of adeG complementation to restore the phenotype contradicts the claim that aggregation is adeG-specific and instead suggests polarity effects, secondary mutations, or background strain defects. These issues remain unresolved and require further clarification or experimental validation before attributing the aggregation phenotype specifically to adeG.
- Response to Comment 6: The explanation for the opposing adeJ expression patterns remains unconvincing. Because adeIJK is primarily regulated by AdeN (a TetR-type transcriptional regulator), the observed transcriptional changes likely reflect background-specific regulatory differences rather than a direct consequence of adeG YZUMab17 carries extensive RND regulatory defects (ΔadeR, ΔadeABC), whereas ZJab1 does not, making these strains not directly comparable. The current interpretation overstates causality and does not adequately address how adeG deletion mechanistically leads to opposite adeJ expression in distinct backgrounds. Additional evidence or clarification is needed.
Author Response
- Comments on responses- 2, 3 and, 4: Comments on responses- 2, 3 and, 4: Thank you for the clarification. However, the explanation remains biologically inconsistent. In RND efflux systems, loss of the core transporter (AdeG) should disrupt the entire AdeFGH pump; therefore, the ΔadeG phenotype would normally resemble or intensify the ΔadeFGH phenotype, not differ from it. It is unclear how ΔadeG alone induces aggregation while the complete operon deletion does not.
The proposed idea of AdeF/AdeH forming a “partial complex” with other RND systems (e.g., AdeIJK) is not supported by current understanding, as RND components do not typically mix across operons.
Additionally, the failure of adeG complementation to restore the phenotype contradicts the claim that aggregation is adeG-specific and instead suggests polarity effects, secondary mutations, or background strain defects. These issues remain unresolved and require further clarification or experimental validation before attributing the aggregation phenotype specifically to adeG.
Response: We highly appreciate the reviewer’s insightful and professional guidance, which has prompted us to conduct a more in-depth analysis of our study. We fully agree with the reviewer’s perspective that the differential aggregation phenotypes of ΔadeG across strains cannot be solely and definitively attributed to the specific regulatory role of adeG in aggregation. Instead, these phenomena are likely caused by the combined or individual effects of polarity effects, secondary mutations, or background strain defects. To address this concern, we have removed all definitive statements regarding the mechanisms by which adeG regulates aggregation from the manuscript. In the Discussion section, we have added speculative interpretations regarding the potential causes underlying the aggregation phenotype induced by adeG deletion in certain strains. (Line 1026-1036)
Furthermore, we plan to verify and clarify the exact mechanisms in future studies by performing whole-genome sequencing of wild-type strains, ΔadeG mutants, and complemented strains, combined with targeted gene editing approaches. These experiments will help determine whether adeG deletion triggers off-target mutations, interferes with the expression of upstream/downstream genes, or results in functional defects such as intrinsic regulatory pathway abnormalities, gene polymorphisms, or recessive mutations, all of which may contribute to the observed aggregation. Given that these follow-up studies require additional time to complete, and crucially, that the outcomes of these experiments will not affect the core conclusion of this manuscript—aggregation of planktonic Acinetobacter baumannii significantly enhances antibiotic tolerance and virulence—we intend to report the results of these subsequent investigations as part of future research projects.
- Response to Comment 6: The explanation for the opposing adeJ expression patterns remains unconvincing. Because adeIJK is primarily regulated by AdeN (a TetR-type transcriptional regulator), the observed transcriptional changes likely reflect background-specific regulatory differences rather than a direct consequence of adeG YZUMab17 carries extensive RND regulatory defects (ΔadeR, ΔadeABC), whereas ZJab1 does not, making these strains not directly comparable. The current interpretation overstates causality and does not adequately address how adeG deletion mechanistically leads to opposite adeJ expression in distinct backgrounds. Additional evidence or clarification is needed.
Response: We fully concur with the reviewer’s comment. We have revised the overstated causal interpretations in the original manuscript and now clearly state that strains YZUMab17 and ZJab1 exhibit substantial genetic background differences. The observed opposing transcriptional patterns of adeJ are therefore more likely to reflect background-specific regulatory variations rather than a direct regulatory effect of adeG deletion. (Line 1037-1049)
Reviewer 2 Report
Comments and Suggestions for Authors
The authors have adequately addressed my comments.
Author Response
Comments: The authors have adequately addressed my comments.
Response: We sincerely appreciate the reviewer for your thorough review and positive feedback. Your valuable comments have been instrumental in refining the scientific rigor and clarity of our manuscript. We are pleased to know that our revisions have adequately addressed all your concerns, and we thank you again for your time, expertise, and constructive guidance throughout the review process.